

# Sensitivity, stability and future evolution of the world's northernmost ice cap, Hans Tausen Iskappe (Greenland)

Harry Zekollari[1,2], Philippe Huybrechts[1], Brice Noël[3], Willem Jan van de Berg[3] and Michiel R. van den Broeke[3]

[1] Earth System Science & Departement Geografie, Vrije Universiteit Brussel, Brussels, Belgium
[2] Laboratory of Hydraulics, Hydrology and Glaciology (VAW), ETH Zürich, Zurich, Switzerland
[3] Institute for Marine and Atmospheric Research, Universiteit Utrecht, Utrecht, The Netherlands

*Correspondence to*: Harry Zekollari (harry.zekollari@vub.ac.be)

**Abstract.** In this study the dynamics and sensitivity to climatic forcing of Hans Tausen Iskappe (western Peary Land, Greenland) are investigated with a coupled ice flow – mass balance model. The surface mass balance is calculated from a Positive Degree-Day runoff/retention model, for which the input parameters are derived from field observations. The precipitation field is obtained from the Regional Climate Model RACMO2.3. For the ice flow a 3-D higher-order thermo-mechanical model is used, which is run at a 250 m resolution. A higher-order solution is needed to accurately represent the ice flow in the outlet glaciers. Compared to the Shallow-Ice Approximation this modifies the steady state ice cap volume by 6-8% and the area by 2-4%. Under 1961-1990 climatic conditions a steady state ice cap is obtained that is overall similar in geometry to the present-day ice cap. Ice thickness, temperature and flow velocity in the interior agree well with observations. For the outlet glaciers a reasonable agreement with temperature and ice thickness measurements can be obtained with an additional heat source related to infiltrating meltwater. The simulations indicate that the SMB-elevation feedback has a major effect on the ice cap response time and stability. This causes the southern part of the ice cap to be extremely sensitive to a change in climatic conditions and leads to thresholds in the ice cap evolution. Under constant 2005-2014 climatic conditions the entire southern part of the ice cap cannot be sustained and the ice cap loses about 80% of its present-day volume. The projected loss of surrounding permanent sea-ice and corresponding precipitation increase may attenuate the future mass loss, but will be insufficient to preserve the present-day ice cap for most scenarios. In a warmer and wetter climate the ice margin will retreat while the interior is projected to grow, leading to a steeper ice cap, in line with the present-day observed trends. For intermediate (+4°C) and high warming scenarios (+8°C) the ice cap is projected to disappear respectively around 2400 and 2200 A.D., almost irrespective of the projected precipitation regime and the simulated present-day geometry.



## 1 Introduction

Glaciers and ice caps (GICs) made an important contribution to sea level rise in the 20[th] century. Also in the 21[st] century they are projected to be major contributors (Gardner et al., 2011; Jacob et al., 2012; Church et al., 2013; Gregory et al., 2013). The Greenland GICs are no exception as they have contributed significantly in the recent past (Bolch et al., 2013) and will

continue to do so in the coming decades (Machguth et al., 2013; Huss and Hock, 2015). To estimate the magnitude of this regional and global contribution, simplified models have been applied at the regional and global scale (e.g. Marzeion et al., 2012; Slangen et al., 2012; Giesen and Oerlemans, 2013; Radić et al., 2014; Clarke et al., 2015; Huss and Hock, 2015). In order to improve the many parameterizations on which these models rely and for a better understanding of the dynamics of GICs in a changing climate, in-depth modelling studies are needed in which detailed models are used at a high spatial

resolution. A variety of recent studies exist for individual glaciers (e.g. Le Meur and Vincent, 2003; Jouvet et al., 2009, 2011; Aðalgeirsdóttir et al., 2011; Duan et al., 2012; Zekollari et al., 2013, 2014; Hannesdóttir et al., 2015; Réveillet et al., 2015), but for ice caps such detailed studies are limited (Aðalgeirsdóttir et al., 2005, 2006, Flowers et al., 2005, 2007, 2008; Giesen and Oerlemans, 2010). Because of their fundamentally different behaviour, parameterizations developed for mountain glaciers are not valid for ice caps. Whereas a glacier can retreat up the mountain and re-adjust to the new climatic

conditions, an ice cap is unable to do so. Glaciers derive their highest elevation from the surrounding topography, while an ice cap is (largely) self-sustained due to its own height. A decrease in height leads to a decrease in surface mass balance (SMB), a process that can reinforce itself due to a positive feedback and lead to fast collapse. To exactly understand these mechanisms and to explore the stability of ice caps and possible thresholds in the system related to the SMB – elevation feedback, it is necessary to couple the SMB with the ice flow.

Pioneering work on the 3-D modelling of ice caps was undertaken by Mahaffy (1976) who modelled the dynamics of Barnes Ice Cap and compared observed and modelled ice cap geometries. In this study an equilibrium ice cap with a similar size as observed could not be obtained and the ice cap grew far beyond the present-day state or evolved to a very small ice cap. Since then several modelling studies were performed on individual ice caps in Iceland (Aðalgeirsdóttir et al., 2005, 2006,

Flowers et al., 2005, 2007, 2008) and on Hardangerjøkulen (southern Norway) (Giesen and Oerlemans, 2010; Åkesson et al., 2016) with models based on the Shallow-Ice Approximation (SIA). Schäfer et al. (2015) used a Full-Stokes ice flow model with an englacial temperature parameterization for a study on Vestfonna ice cap (Svalbard) with a particular focus on the SMB – elevation feedback. In a recent study Ziemen et al. (2016) modelled the temporal evolution of the Juneau ice field (Alaska), which exhibits some similarities to ice caps. To date, however, no detailed time-dependent thermo-mechanical

modelling studies exist on individual high Arctic ice caps, although these ice caps are projected to be important contributors to sea level rise in the coming decades to century (e.g. Giesen and Oerlemans, 2013; Radić et al., 2014). This contribution is largely driven by an above-average rise in high Arctic temperatures due to the polar amplification (Masson-Delmotte et al.,




2006; Bekryaev et al., 2010; Khan et al., 2014; Lee, 2014; Pithan and Mauritsen, 2014; Seneviratne et al., 2016), which could eventually be attenuated by an increased precipitation (Machguth et al., 2013).

Here we present a 3-D modelling study on Hans Tausen Iskappe (Peary Land, Greenland), the world's northernmost ice cap located between 82.2°N and 83.0°N (Figure 1). Despite its remoteness, a considerable body of field data exists that can be used for model calibration and validation, such as observations on surface mass balance, ice thickness, elevation changes, ice temperature and surface velocity. There are indications that Hans Tausen Iskappe is very sensitive to changes in climatic conditions. Several palaeo-records suggest that after being connected to the Greenland Ice Sheet (GrIS) during the Last Glacial Maximum (LGM) (Bennike, 1987; Larsen et al., 2010), the ice cap (largely) disappeared during the Holocene Thermal Maximum (HTM), after which it started to rebuild around 3500-4500 cal BP (Hammer et al., 2001; Madsen and Thorsteinsson, 2001). In this study we investigate the sensitivity and dynamics of the ice cap and analyse the feedback mechanisms that can lead to fast changes and thresholds in the ice cap evolution. For this purpose we use a coupled SMB - ice flow model at a high horizontal resolution (250 m). In order to resolve the ice flow in the many outlet glaciers accurately, a higher-order (HO) approximation to the full force balance is used. This differs from other detailed ice cap and ice field modelling studies (Aðalgeirsdóttir et al., 2005, 2006, Flowers et al., 2005, 2007, 2008; Giesen and Oerlemans, 2010; Hannesdóttir et al., 2015; Åkesson et al., 2016; Ziemen et al., 2016) that are based on the Shallow-Ice Approximation (SIA) or similar approaches. We investigate the influence of the model complexity (SIA/HO) and resolution on the modelled geometries and run the ice cap into a steady state that is compared to field observations. At first the thermo-mechanical ice-flow model (section 3) and SMB model (section 4) are described, after which the model is extensively tuned and validated (section 5). Subsequently thresholds are analysed that could inhibit ice cap growth or decay, followed by the sensitivity of the ice cap to changes in climatic conditions and their implications on the future ice cap evolution (section 6).

## 2 Site description and field data

Hans Tausen Iskappe is an ice cap located in western Peary Land and is separated from the GrIS by the Wandel Dal valley that is 10 to 20 km wide (Weidick, 2001) (see Figure 1). With an area of around 4000 km² (ca. 75 km from north to south and 50 km from west to east) (Starzer and Reeh, 2001) it is the second largest ice cap in northern Greenland after Flade Isblink (ca. 8500 km²) (Kelly and Lowell, 2009; Rinne et al., 2011). It corresponds to around 4-5% of the total area for all GICs in Greenland (ca. 90000 km² (Rastner et al., 2012)), and around 0.5% of the worldwide GICs area. The ice cap has a typical elevation of 1000-1200 m a.s.l., except for local domes that reach up to 1200-1300 m a.s.l.. The outlet glaciers, which are mostly land terminating, often terminate up to several hundred meters above sea level. Some calving glaciers exist, but overall their activity is limited and all fjords have a semi-permanent ice cover which melts only at rare intervals (> 30 years) (Higgins, 1990; Weidick, 2001; Möller et al., 2010). Hans Tausen Iskappe largely covers the underlying topography and therefore qualifies as an ice cap, while other ice masses in Peary Land are smaller and more controlled by the surrounding





topography and therefore rather qualify as valley glaciers or ice fields (e.g. Bure Iskappe, Heimdal Iskappe, Heinrich Wild Iskappe and other ice masses to the north) (Weidick, 2001).

The first documented observations on and around Hans Tausen Iskappe date from the first half of the 20[th] century (Koch,
1928, 1940) and the mid 20[th] century (Davies and Krinsley, 1962), but the first detailed field campaign only occurred in 1975-1976 when several shallow ice and firn cores were drilled. In 1978 an aerial photography campaign was conducted (Starzer and Reeh, 2001), while in the 1990s an elaborate field campaign was set up during the three summers of 1993, 1994 and 1995 (Hammer, 2001). In 1993 different reconnaissance flights were made. Extensive airborne ice thickness measurements were performed from Twin-Otter aircrafts (Thomsen et al., 1996; Starzer and Reeh, 2001). Additional
ground-based ice thickness measurements were performed in 1994 and 1995 at a variety of locations (Gundestrup et al., 2001). From these measurements it is known that the ice cap rests on a 800-1100 m a.s.l. elevated plateau in the northern part, while in the southern part the bedrock elevation is lower and varies between 600 to 900 m a.s.l. (Figure 2a). Consequently in the southern part the ice is substantially thicker and locally reaches more than 500 m along a north-south oriented deep canyon (Gundestrup et al., 2001) (Figure 2b). Based on an interpolation of these measurements the ice cap had
an estimated volume of around 760 km[3] in the mid 1990s (Starzer and Reeh, 2001). Notice that the direct ice thickness measurements on Hans Tausen from the 1990s are not included in the Bamber et al. (2013) dataset and therefore some inconsistencies exist with the reconstructed bedrock from Starzer and Reeh (2001). The surface elevation in both datasets is largely similar and only small discrepancies exist that may partly be linked to the different time of acquisition.

From 1994 to 1995 a strain network was set up around the Central Dome, which is sometimes referred to as 'the southeastern dome' (Reeh, 1995; Gundestrup et al., 2001; Hvidberg et al., 2001; Jonsson, 2001). A mass balance measurement programme was established between North Dome (1320 m a.s.l.) and Hare glacier, a small outlet glacier with its front at 220 m a.s.l. (see Figure 1) (Reeh et al., 2001; Machguth et al., 2016). Here different components of the energy balance were measured and a detailed stake farm was set up (Braithwaite et al., 1995). In 1995 a 345-m ice core was drilled at Central
Dome (1271 m a.s.l.) (Hammer et al., 2001; Johnsen et al., 2007), which combined with glacial geological investigations (Landvik et al., 2001) provide constraints on the palaeo ice cap evolution. Additionally englacial temperatures (Reeh, 1995; Thomsen et al., 1996) and surface velocities (Reeh, 1995; Hvidberg et al., 2001) were measured and were used for model tuning and validation in this study.

Surface velocities are known from Interferometric Synthetic Aperture Radar (InSAR) measurements for the winters of 2000/1, 2005/6, 2006/7, 2007/8, 2008/9 and 2009/10 (Joughin et al., 2010, 2015). These are around a few meters per year for the interior, around 30-60 m a[-1] for medium sized outlet glaciers (e.g. Hare glacier), and up to 200 m a[-1] for the largest outlet glaciers (Figure 8a). These values are in agreement with the surface velocities derived from the mass balance stakes on Hare glacier (Thomsen et al., 1996) and a strain network setup around the Central Dome (Hvidberg et al., 2001). The InSAR





velocities are consistent with the 1947-1978 average surface velocities deduced from aerial photography (Higgins, 1990), although direct comparisons are not always straightforward as the exact location of the aerial measurement is often not clearly stated (see Joughin et al. (2010) for an elaborate discussion).

**3 Thermo-mechanical ice flow model**

**3.1 Ice flow model and experimental setup**

Nye's Generalization of Glen's flow law is used as a constitutive equation for ice deformation (Glen, 1955; Nye, 1957), where the deviatoric stresses ($\tau_{ij}$) are defined as:

$$\tau_{ij} = 2\eta \dot{\varepsilon}_{ij} \tag{1}$$

$$\eta = \frac{1}{2} A(t)^{-1/n} (\dot{\varepsilon}_e + \dot{\varepsilon}_0)^{\frac{1}{n}-1} \tag{2}$$

Here $\eta$ is the viscosity, $n$ is the power-law exponent (set to 3), $A(t)$ is the rate factor, which is temperature dependent following an Arrhenius type function, and $\dot{\varepsilon}_0$ is a small offset ($10^{-30}$) that ensures finite viscosity (Fürst et al., 2011). The

effective stress $\dot{\varepsilon}_e$ is determined from the second invariant of the strain-rate tensor:

$$\dot{\varepsilon}_e^2 = \frac{1}{2} \dot{\varepsilon}_{ij} \dot{\varepsilon}_{ij} \tag{3}$$

where the strain tensor $\dot{\varepsilon}_{ij}$ is defined as:

$$\dot{\varepsilon}_{ij} = \frac{1}{2} (\partial_i u_j + \partial_j u_i) \tag{4}$$

A widely used approximation for ice sheet and ice cap modelling is the Shallow Ice Approximation (SIA) in which only the local shear stresses are accounted for and the longitudinal components are neglected (Hutter, 1983). Under the SIA equations (1-4) are simplified and the shear stress results from vertical plane shearing:

$$\partial_i \tau_{iz} = \rho g \partial_i s \tag{5}$$

Here $\rho$ is the ice density, $g$ is the gravitational acceleration and $s$ is the surface elevation. The SIA approximation is based on a large width-depth ratio and is valid for the interior of the ice cap, but not for its many narrow outlet glaciers. To more accurately represent the ice flow in the outlet glaciers a higher-order (HO) approximation to the Stokes momentum balance is therefore used in which longitudinal stress components are accounted for (Blatter, 1995; Pattyn, 2003; Fürst et al., 2011). More specifically a multilayer longitudinal stresses approximation of the force balance, abbreviated as LMLa in Hindmarsh

(2004), is used, where a cryostatic equilibrium in the vertical is assumed by neglecting bridging effects (i.e. neglecting vertical resistive stresses):

$$\partial_i (2\tau_{ii} + \tau_{jj}) + \partial_j \tau_{ij} + \partial_z \tau_{iz} = \rho g \partial_i s \qquad \text{(for } i \neq j\text{)} \tag{6}$$

$$\dot{\varepsilon}_{iz} = \frac{1}{2} \partial_z u_i \tag{7}$$





This HO approximation and its numerical implementation (Fürst et al., 2011) have been successfully applied at different scales, ranging from small mountain glaciers (Zekollari et al., 2013, 2014; Zekollari and Huybrechts, 2015) to entire ice sheets (Fürst et al., 2013, 2015).

All ice-free patches located within the present-day ice cap, which mainly coincide with mountain peaks and steep ridges, are explicitly kept ice free in the simulations, as the processes that prevent accumulation here (mostly snowdrift by wind and related to the steep topography) are not captured in our models. To prevent confluence of ice flow from nearby small ice masses (mainly from Bure Iskappe and Heimdal Iskappe in the east) and from the GrIS, which only partly belong to the domain and can therefore not be modelled explicitly, ice is only allowed to grow from areas that are covered by Hans Tausen

Iskappe at present. The ice can subsequently expand freely, without any constraints (e.g. can connect to the GrIS), except for areas where the bedrock elevation is lower than -50 m, where the ice is removed as a crude representation of calving. Field and aerial observations (Weidick, 2001) suggest that calving is very limited (up to a 2-3% of total mass loss) in occurrence and magnitude and this is also the case in our numerical simulations.

### 3.2 Thermodynamics and role of meltwater

A full 3-D calculation of the ice temperature is performed simultaneously with the velocity calculations as the ice temperature relates to the ice stiffness (rate factor in Glen's flow law) and determines whether or not basal sliding occurs (see e.g. Huybrechts (1996) for a more detailed account). Surface temperature is calculated from the mean annual temperature and a warming component related to the superimposed ice formation. Observed refreezing of slush fields (Reeh, 1995), (sub/)surface temperature and surface isotopes measurements (Thomsen et al., 1996; Reeh et al., 2001) suggest that

refreezing occurs. Based on field measurements (Reeh et al., 2001) a firn warming of 20 K / m ice equivalent refreezing is used. At the base of the ice cap heat is produced by the geothermal heat flux and friction generated by basal sliding.

In the percolation zone of the ice cap temperature measurements suggest an additional heat source from infiltrating supraglacial meltwater that can reach the bed. We incorporate this additional heat source by imposing a basal-water heat flux

(similar to geothermal heating), following the approach of Wohlleben et al. (2009). This mechanism differs from cryo-hydrologic warming (Phillips et al., 2010), where meltwater heating also plays a crucial role (through flowing, ponding and refreezing), but where the heat source is spread and leads to a warming of the whole ice column. A heating through meltwater is supported by high values from englacial temperature measurements in the percolation zone of the ice cap (Thomsen et al., 1996) and is also observed on the GrIS (Thomsen et al., 1991; Phillips et al., 2010, 2013; Lüthi et al., 2015).

There are also observations on other Arctic ice caps, such as Laika ice cap (Canada), where repeatedly measured high englacial temperatures cannot be explained without an additional heat source (Blatter and Kappenberger, 1988; Blatter and Hutter, 1991), and a 200-m thick ice column in the ablation zone of Barnes ice cap where the 10-m depth temperature is -10°C, while temperatures at 130 m depth are close to the pressure melting point (Classen, 1977). In a recent study on the



Qaanaaq ice cap (NW Greenland), Sugiyama et al. (2014) indicate that the observed velocities cannot be reproduced without accounting for a heat transfer from meltwater when solving for thermodynamics, and Schäfer et al. (2014) also stress the possible effect of meltwater on thermodynamics and ice flow for Vestfonna ice cap (Svalbard).

## 4 Surface mass balance (SMB)

The SMB model used in this study calculates melt and runoff from the widely used Positive Degree-Day (PDD) runoff/retention approach (Reeh, 1989; Janssens and Huybrechts, 2000; Gregory and Huybrechts, 2006). This approach allows for the SMB to be calculated at any time for any geometry. In this study the SMB model is coupled to the thermo-mechanical ice flow model once a year, avoiding potential problems related to a long coupling interval (see Schäfer et al., 2015). The SMB is also available from the RCM RACMO2.3 (Noël et al., 2015) for a fixed geometry, but it is not directly

used as the PDD is more flexible for generating the SMB under an evolving geometry without the need to modify RCM output for a different surface elevation (Franco et al., 2012; Helsen et al., 2012; Edwards et al., 2014). The implications of using such correction methods to account for the interaction between surface elevation and SMB were investigated in detail by Schäfer et al. (2015) for Vestfonna ice cap (Svalbard), suggesting that such approaches should not be used for simulations of more than several decades and under extreme climate change scenarios.

### 4.1 Model setup

#### 4.1.1 Positive Degree-Day approach

The PDD runoff/retention model determines the PDD sum from monthly air temperatures assuming a variability of daily near-surface (2 m) temperatures around the monthly mean. Melt rates are proportional to this melt potential. In snow covered regions the meltwater from surface melting is initially stored as capillary water within the snowpack, until the snowpack

becomes saturated, typically when melt reaches around 60% of the annual precipitation (Janssens and Huybrechts, 2000), and runoff occurs. The formation of superimposed ice occurs when water saturated snow survives above the impermeable ice layer until the end of the season, and subsequently refreezes (Janssens and Huybrechts, 2000).

The variability in temperatures is expressed as a standard deviation ($\sigma$) around the monthly mean. A value of 3°C is used,

lower than the widely used value of 4.2°C for the whole of Greenland (e.g. Fürst et al., 2015), but consistent with 1994-95 observations of temperatures, SMB and refreezing for 6 locations on Hare glacier (Reeh et al., 2001). The melt rates are determined with separate degree-day factors for snow and ice, which are respectively equal to 0.0027 and 0.0065 m ice equivalent / degree-day based on detailed stake observations on the ice cap (Braithwaite et al., 1995; Reeh, 1995).





### 4.1.2 Temperature parameterization

The PDD sum is calculated from a parameterization of the mean annual temperature (TMA) and the mean July temperature (TMJ), assuming a sinusoidal annual march. The mean July temperature (TMJ) has the largest influence on the SMB as this field largely determines the amount of summer melt, while the TMA sets the amplitude of the annual sinusoidal signal and

thereby determines the temperatures of the other seasons, when little to no melt occurs. The rain-snow temperature threshold is set at 1°C. TMA and TMJ are parameterized for the mass balance year 1994/95 based on detailed field measurements (Reeh, 1995; Reeh et al., 2001) and this parameterization is subsequently extended to the period 1961-1990 by comparing both periods in the regional climate model RACMO2.3 (with ERA-40 and ERA-interim boundary conditions): the 1994-95 TMA was 0.53°C higher than the 1961-1990 average, while the 1995 TMJ was about 0.27°C lower than the 1961-1990

average.

TMJ is parameterized as a function of latitude and elevation. The measured temperature lapse rate in July is -0.0056°C / m (Reeh et al., 2001) and as a latitudinal gradient we adopt the value of -0.1681°C / °N from Fausto and others (2009). Altogether this results in the following parameterization for the 1961-1990 mean July temperature (in °C) (see Figure 3a):

$$TMJ = 19.47 - 0.1681 \times LAT - 0.0056 \times ELEV \qquad (9)$$

Where LAT is the latitude (°) and ELEV the elevation (m). Notice that given the limited latitudinal range of the ice cap (0.8°) the influence of the latitudinal gradient is limited. The parameterized 1961-1990 mean July temperature (TMJ) agrees well with output from RACMO2.3 for the period 1961-1990 (Figure 3b), with mean July temperatures around 5.5-6°C at sea level and around -1 to -1.5°C at the domes (1200-1300 m a.s.l.).

The mean annual temperature has not been measured on the ice cap directly. The latitudinal and elevation gradients are therefore also taken from Fausto and others (2009). A parameterization is derived to fit with measurements from the nearby meteorological station Kap Harald Moltke (Reeh et al., 2001). For areas lower than 300 m, no elevation gradient is applied in order to represent the temperature inversion that occurs here following Reeh et al. (2001). For 1961-1990 the mean annual temperature (TMA) (in °C) is parameterized as follows (see Figure 3c):

$$for\ ELEV < 300\ m: TMA = 45.07 - 0.734 \times LAT$$

$$\qquad (10)$$

$$for\ ELEV > 300 m: TMA = 46.97 - 0.734 \times LAT - 0.00638 \times ELEV$$

With this parameterization the 1961-1990 mean annual temperature (TMA) at sea level is around -15°C, while at North Dome this is -22.2°C and at Central Dome -21.7°C. The TMA at the domes agree well with the local measurements of 10-m depth temperature, which are respectively -21.0°C and -20.8°C (Reeh, 1995; Reeh et al., 2001). The slight difference between parameterised and observed may be linked to firn warming due to refreezing, although this process is generally





limited at those high elevations. Furthermore the parameterized TMA is in fairly good agreement with RACMO2.3 output for this period (see Figure 3d).

### 4.1.3 Precipitation parameterization

The accumulation has been derived from field measurements and several shallow cores that cover most of the 20[th] century

(Reeh et al., 2001). The accumulation is the highest in the north(west), which is related to the proximity to the ocean acting as a moisture source (see Figure 3e). Due to a topographic shielding the precipitation is much lower in the central part. In the south(east) the precipitation is also very low, but generally a bit higher than in the centre. Based on accumulation measurements from shallow cores Reeh et al. (2001) tried to derive a parameterization to describe this pattern as a function of latitude and elevation, but this resulted in an unrealistic field with very low (in some cases even negative) precipitation at

low elevation. We therefore opt to use the precipitation field from the regional climate model simulation RACMO2.3, which is run at 11 km and was bi-linearly downscaled to a 1 km resolution (see Figure 3e for 1961-1990 average field) (Noël et al., 2016). This regional climate model is able to reproduce the observed precipitation patterns closely as is shown by a comparison with accumulation from 4 sites for the period 1975-1995 (**Table 1**). Field observations, and model output (both from our SMB model and from RACMO2.3) indicate that for the 4 high-elevation sites the solid precipitation is nearly equal

to the accumulation (annual rain fraction varies between 0 and 5%).

### 4.2 SMB model evaluation

The SMB model is applied to the observed geometry (Starzer and Reeh, 2001) for the mass balance year 1994/95 and compared to the measured SMB. For Hare outlet glacier (see Figure 1), where an extensive mass balance network was installed (Braithwaite et al., 1995; Reeh et al., 2001), the modelled SMB agrees with the observations. For this relatively wet

year (Reeh et al., 2001) for Hare glacier the frontal SMB is below -1 m w.e. a$^{-1}$, the Equilibrium Line Altitude (ELA) is around 700-750 m and the highest SMB is around 0.3 m i.e. a$^{-1}$ at 1300 m (Thomsen et al., 1996; Machguth et al., 2016) (**Figure 4**a). SMB measurements in the ablation area occurred between 25/07/1994 and 17/08/1995 and mainly reflect the 1994-95 SMB and some late summer melt from the 1993-94 balance year. They should therefore be considered as an underestimation (lower bound) for the 1994/95 SMB (**Figure 4**a). The SMB measurements in the accumulation area span the

period between 04/08/1994, after which the local melt is very limited, and 23/06/1995, before which melt is limited, and are therefore close to the 1994/95 SMB (**Figure 4**a).

For the period 1961-1990 the average SMB from our PDD runoff/retention model is compared to the 1-km downscaled SMB version v1.0 from the regional climate model RACMO2.3, which is reconstructed by adding up daily-downscaled runoff,

sublimation, snowdrift erosion and total precipitation (rain and snow) (Noël et al., 2016) (**Figure 4**). For this comparison the SMB is modelled using the same 1-km GIMP DEM ice mask and topography (Howat et al., 2014) for both models. The output from both models is in relatively good agreement (see **Figure 4**b), which is in part related to the fact that the





precipitation forcing is the same in both models. For both approaches the integrated 1961-1990 SMB over the ice cap is close to zero: -0.02 m w.e a$^{-1}$ in RACMO2.3 and +0.02 m w.e. a$^{-1}$ in the PDD approach. The slightly lower SMB in RACMO2.3 results from a stronger melt/sublimation component, due to which the ELA, especially for the southern part of the ice cap, is lower compared to the PDD approach (see **Figure 4**c,d). As a result the RACMO2.3 SMB field has a more

pronounced imprint of the elevation field, while the PDD SMB field has a stronger imprint from the precipitation field. Reconstructions for other periods, that are addressed further, show that both approaches are also in generally good agreement for other periods in time, which is in line with an earlier study by Hanna et al. (2011). They compared the PDD approach and RACMO2.1 output for Greenland for the period 1958-2010 in terms of interannual variability and found a reasonable agreement.

**5 Ice cap under 1961-1990 climatic conditions**

**5.1 1961-1990 modelled steady state (250 m, HO) vs. observations**

The 1961-1990 SMB conditions are imposed and the ice cap is run into a steady state using the coupled HO thermo-mechanical – PDD model, which is run at a 250 m horizontal resolution.

**5.1.1 Ice cap extent and SMB**

The steady state ice cap obtained from the 250-m HO run (Figure 5) is close in extent to the observed ice cap in 1995 (Figure 2). A few discrepancies exist in the south-western part, where the steady state ice cap is somewhat smaller in extent, and for a few outlet glaciers in the central northern part, which are a bit shorter than in the observations. Notice that the latter outlet glaciers are very thin (Starzer and Reeh, 2001) and satellite derived surface velocities indicate that these areas are almost stagnant (Joughin et al., 2010, 2015). In the south-eastern and north-western part the modelled steady state ice cap extends

slightly further than the observations and some of the present-day ice-free ridges between the outlet glaciers are ice-covered, but except that also here the agreement is overall relatively good, especially given that there is no imposed constraint on the ice cap extent.

The modelled limited areal changes under the 1961-1990 average conditions are supported by output from RCMs that

indicate that the average integrated SMB over Hans Tausen for this period is close to zero. Using ERA40 (1958-1978) and ERA interim (1979-1990) as boundary conditions, the average SMB in RACMO2.3 (11 km run, downscaled to the 1 km GIMP DEM, see Sect. 4.2) is -0.02 m w.e. a$^{-1}$ (see **Figure 4**b). In another widely used RCM for Greenland, MAR3.5.2 (20 km run, downscaled to the 5 km Bamber et al. (2013) DEM) (Fettweis et al., 2013) an integrated SMB of +0.03 m w.e. a$^{-1}$ is obtained. Given the different topographic input a direct comparison between both RCMs is difficult, but both indicate a near

zero SMB for this period. This is also the case from our PDD melt/retention model, where the average 1961-1990 SMB is +0.02 m w.e. a$^{-1}$ based on the 1-km GIMP surface elevation (Howat et al., 2014) (see **Figure 4**a), or +0.03 m w.e. a$^{-1}$ when





using the reconstructed geometry from Starzer and Reeh (2001) as an input. Furthermore the limited geometrical changes under the 1961-1990 climatic conditions are also in line with field evidence. After reaching a maximum extent around 1900, which is known from Little Ice Age (LIA) moraines (Koch, 1928, 1940), and slightly retreating in the first part of the 20[th] century (Davies and Krinsley, 1962), aerial photography from the 1970s and 1990s (Weidick, 2001) suggest that the second

part of the 20[th] century is characterized by a slower recession (limited to tens of meters), stand-still or even slight readvances. A recent study by Kjeldsen et al. (2015), where aerial photography and SMB modelling are combined, also suggests limited mass changes in northern Greenland for the period 1900-1983 and an ice sheet near balance during the 1970s and 1980s.

### 5.1.2 Englacial ice temperatures

The measured temperature profiles at Central Dome and at Hare glacier (Reeh, 1995; Thomsen et al., 1996) are used to tune

the geothermal heating component and the heating component related to infiltrating meltwater in the ablation area (Figure 6). To reproduce the observed englacial temperatures at Central Dome (Reeh, 1995) (see Figure 6 c), a geothermal heat flux of 45 mW m$^{-2}$ is applied. Here the modelled steady state ice thickness (318 m) is close to the observed one (345 m). The measured almost linear decrease in temperature, from -21°C at 10 m depth (-21.7°C in our model, corresponding to the TMA at the surface) to -16°C, are closely reproduced. A geothermal heat flux of 45 mW m$^{-2}$ is lower than the 60 mW m$^{-2}$

interpolated from Shapiro and Ritzwoller (2002) to the location of Hans Tausen Iskappe. However with 60 mW m$^{-2}$ the modelled basal temperature is equal to –13.8°C and the local ice thickness is 295 m. For the ablation area an additional basal-water heat flux of 150 mW m$^{-2}$ is adopted to reproduce the englacial temperatures measured in the ablation area of Hare glacier (Reeh, 1995). With this additional basal heating the observed temperatures, ranging from -18.5°C (at 10 m depth) to about -1.5°C at the bottom (see Figure 6), are well reproduced. Despite the fact that the modelled ice thickness (269

m) is close to the measured one (289 m) a direct comparison is difficult to make, as the imposed surface temperature (–15.3°C, corresponding to the local TMA) is slightly higher than the observed one. With a value of 150 mW m$^{-2}$ the highest basal temperatures for Hare glacier are close to the pressure melting point (see Figure 6b), but this point is nowhere reached and therefore basal sliding does not occur here. The pressure melting point is only reached for a few larger outlet glaciers and only very locally (see Figure 6a) and the modelled contribution of basal sliding is therefore very limited. Our value of

150 mW m$^{-2}$ differs significantly from the the 350 mW m$^{-2}$ found by Wohlleben et al. (2009), which is expected given the very different setting (location, SMB, meltwater production and infiltration mechanisms) and the different methodological approach. Here we tune based on an evolving/modelled geometry, while Wohlleben et al. (2009) model the thermodynamics for a fixed geometry. With a value of 350 mW m$^{-2}$ almost the entire ablation area of the ice cap would be at the pressure melting point and basal sliding would have an important role, which is not supported by the field evidence. Notice that on

the other hand, without any additional basal heating component for the ablation area, the basal temperatures would be severely underestimated (e.g. -10.2°C at the base of the Hare glacier drill site), which would also strongly affect the ice cap geometry.



### 5.1.3 Ice cap geometry

The 1961-1990 modelled steady state geometry is generally in good agreement with the observed geometry. The observed ice thickness (Figure 2b) is well reproduced in the model (Figure 5) and so is the surface elevation as the bedrock elevation is the same in the observations and the model. The Root Mean Square Error (RMSE) between the observed and modelled ice

thickness (and surface elevation) is 55.6 m. For the interior of the ice cap the regions with high ice thickness are generally well reproduced, despite some differences in the North, where the ice cap is generally thicker in the model. The steady state outlet glaciers agree reasonably well with observations (RMSE of 67.1 m), but some modelled outlet glaciers, especially in the North, have a tendency to be slightly thicker. This difference is partly linked to the complexity of ice flow in the outlet glaciers, which may not be fully captured by the model. The ice flow in the outlet glaciers is strongly influenced by

thermodynamics as the ice temperature determines the stiffness, through the rate factor, and potentially also through basal sliding. Without the additional heat source in the ablation area, which was needed to reproduce the observed temperatures, the modelled outlet glaciers are thicker and the discrepancy between observations and model substantially increases (RMSE of 74.7 m for the ablation area).

### 5.1.4 Surface velocities

The surface velocity patterns derived from InSAR data (Joughin et al., 2010, 2015) are well reproduced in the modelled steady state ice cap (Figure 7). The low velocities in the interior and the ice flow direction are very similar, which indicates that the modelled position of the ice divides corresponds well to the observations. For the outlet glaciers, many of the observed velocity patterns are closely reproduced, as is illustrated for a few of the main outlet glaciers at the eastern side of the ice cap (Figure 7c,d), for which the modelled geometry is in relatively good agreement with the observations (see Figure

2b and Figure 5).

The main differences between the modelled and observed ice velocities occur along the south-western edge of the ice cap, where the geometrical differences are the largest, and for two high velocity floating tongues, which we do not model explicitly as this is of limited importance to the larger scale dynamics of the ice cap. For a few outlet glaciers in the North

the modelled surface velocities are slightly higher than the observed ones, which can partly be linked to the differences in geometry.

### 5.1.5 Steady state and implications

Given its long response time, the ice cap in 1995 is not expected to be in steady state with the 1961-1990 conditions. Our model simulations however suggest that the ice cap changes only little under these conditions, and this is supported by field

and SMB modelling evidence, but in reality the ice cap cannot have been in a full dynamic equilibrium with these climatic conditions. In order to calibrate (for thermodynamics) and validate (geometry, surface velocities) our model we however





need to rely on a steady state geometry. A part of the described differences between observations and modelling are therefore not only linked to the model errors and uncertainties in the input data, but may also be attributed to the fact that the observed ice cap was not in steady state in 1995. This is particularly the case for the outlet glaciers, which are the most dynamic parts of the ice cap and where the model-observation discrepancies in geometry and surface velocities are the largest. In order to
fully investigate the transient behaviour of the ice cap, which would be needed to reproduce the recently observed changes and to make accurate projections for the near future (coming decades), simulations encompassing the long-term ice cap evolution are needed. This is however not the focus of this study as we are aiming to understand the large-scale dynamics, response time and climatic sensitivity of this ice cap, and their implications for its long-term evolution.

### 5.2 Impact of horizontal resolution and model complexity

In order to analyse the impact of the horizontal resolution the model is also run at a 500 m resolution. For both runs (250 m and 500 m) the SMB is calculated based on the 1961-1990 climatology applied on the present-day geometry and fixed in time in order to make a 'clean' comparison and avoid effects related to the SMB-elevation feedback. The main differences occur for the narrow outlet glaciers (Figure 8a), which are typically only a few kilometres wide and which are therefore difficult to accurately represent at a 500 m resolution as they only encompass a few grid cells at this resolution. At a 250 m
resolution the modelled surface velocities are generally higher compared to the 500 m run (see Figure 7d,e), which leads to a slightly lower local ice thickness in the outlet glaciers (see Figure 8a). For a coupled run, where the SMB is linked to the evolving geometry, this results in a 3% higher volume for the 500-m run, which translates into a 1% larger area due to the SMB-elevation feedback as the integrated mass balance of the ice cap needs to be zero for it to be in steady state. Notice that treatment of the ice mask in the downscaling approach has an important effect on the modelled geometry at a 250 m
resolution and that it is important that the area of the ice cap and ice free regions is the same at both resolutions in order to ensure that the large-scale dynamics, which are determined by the overall mass balance, are similar.

The effect of a change in model complexity (SIA vs. HO) is also mostly visible in the outlet glaciers (Figure 8b). Whereas the SIA is a local solution, which depends on the local ice thickness and surface slope, the HO solution accounts for the
longitudinal stress gradients, which result in a smoothing of the velocity field (i.e. non-local solution) (cf. Fürst et al., 2013). As a result the highest velocities, i.e. situated around the ELA, are lower in the HO simulations compared to the SIA simulations (Figure 7d,f) and the SIA surface velocities are overestimated compared to the observations (Figure 7c). This leads to thicker outlet glaciers in the HO solution compared to the SIA. The ice cap steady state volume is 7% higher and as a result the area increases by 2.5%.




## 6 Ice cap stability and sensitivity to climatic forcing

### 6.1 Importance initial conditions

The evolution of the ice cap shows evidence of hysteresis as for certain climatic conditions the final steady state geometry is a function of the initial condition. Four different cases arise depending on the imposed climatic conditions.

In case 1, under conditions colder than -0.2°C or colder compared to the 1961-1990 average climatic conditions, the initial geometry does not influence the final steady state: i.e. whether starting from an ice-free surface or from the 1961-1990 steady state (or from the present-day geometry), the ice cap evolves to the same steady state geometry (see Figure 9a,b; Figure 10). Under these climatic conditions the SMB allows for an ice-cap wide build-up, even when starting from ice-free

conditions.

Case 2 occurs for slightly warmer conditions, for a forcing of -0.2°C to +0.35°C compared to 1961-1990, where the ice cap also evolves to the same steady state, but where the ice supply from the northern to the southern plateau plays a crucial role. When the 1961-1990 average climatic conditions are imposed on an ice-free surface the ice initially builds up on the

northern plateau and at a few isolated locations on the lower-lying southern plateau. On the northern plateau the ice cap quickly builds up and a mass flux to the southern plateau initiates around 2.5-3 ka. This ice flux from the northern to the southern part leads to a colonization of the deep southern canyons and a fast build-up of the southern ice cap occurs as a result of the SMB elevation feedback. This evolution is clear from the volume evolution rate (Figure 9a), which after starting to decrease between 1.5 and 3 ka, remains at a steady level between 3 and 8 ka and finally gradually decreases until a new

steady state is reached. As a consequence of this particular ice supply here the shape of the volume evolution curve is far less exponential compared to case 1 (Figure 9, 1961-1990 - 0.5°C).

Case 3 corresponds to further warming for a temperature forcing between +0.35°C and +0.65°C (vs. 1961-1990). Here, the initial geometry will influence the final steady state, i.e. a hysteresis occurs (Figure 10). This is the case for the evolution

under the 1981-2010 conditions, which according to the RACMO2.3 simulations are around 0.6°C warmer than the 1961-1990 average conditions. Under these conditions and starting from an ice-free surface the ice flow from the northern plateau to the southern lower lying areas is insufficient and as a consequence the southern part of the present-day ice cap cannot start to grow due to the SMB–elevation feedback (Figure 9d). The build-up of the ice cap is much faster than under the 1961-1990 conditions as there is no interaction between the northern and southern part of the ice cap. The volume response time,

defined as the time needed to reach $1-e^{-1}$ of the final volume, is in this case 1053 years. Under the same climatic conditions and considering the 1961-1990 steady state ice cap geometry as a starting point, the southern part of the ice cap does not disappear as the SMB is more positive due to the higher elevation (Figure 9c). The existence of this threshold in the system





is therefore strongly related to the particular bedrock geometrical setting, with the high plateau in the north and the ice flow feeding mechanism to the lower lying southern plateau.

In case 4, for even warmer conditions, a warming of more than +0.65°C compared to 1961-1990, the SMB of the southern
part of the present-day ice cap crosses a lower bound, 'the collapse threshold', at which this parts fully disappears because of the SMB-elevation feedback (Figure 9e). Here the ice cap also evolves to a similar steady state (Figure 10) with no ice on the southern plateau, irrespective of the initial condition.

### 6.2 Ice cap sensitivity to climatic forcing and future evolution

### 6.2.1 Sensitivity to temperature changes

As the previous experiments point out, the ice cap is very sensitive to a change in climatic conditions. For a cooling of only 0.5°C compared to the 1961-1990 conditions the ice cap largely expands (see Figure 9b) and the volume increases by 26%. These are the coldest conditions for which the ice cap can be modelled explicitly, as for lower temperatures the ice cap starts to connect to the GrIS and other nearby ice masses and expands beyond the domain boundaries.

Under the 1981-2010 average climatic conditions (ca. +0.6°C vs. 1961-1990, largely similar precipitation pattern) the SMB of the southern part of the present-day ice cap is still above the 'collapse threshold' (cf. case 3, see Figure 9a). While the northern part of the ice cap and the local domes change little, an overall slight decrease in surface elevation occurs at lower elevations and a frontal retreat of the southern part of the ice cap occurs, but then the ice cap quickly stabilizes. This agrees with observations from airborne surveys that indicate that between 1994 and 2004 limited changes in surface elevation
occurred around Central Dome (Dalå et al., 2005). About one fifth of the ice mass is lost under these conditions (Figure 11a). Also the output from the RCMs, MAR3.5.2 (-0.15 m w.e. a$^{-1}$) and RACMO2.3 (-0.11 m w.e. a$^{-1}$), and from our PDD melt/retention approach (-0.08 m w.e. a$^{-1}$ based on the GIMP topography), suggest a limited negative SMB over the ice cap for this period.

For slightly warmer conditions the 'collapse threshold' of the southern part of the ice cap is crossed (cf. case 4, Figure 9a) and eventually, after thousands of years, the entire southern part of the ice cap is lost. This is for instance the case under the 2005-2014 climatic conditions, which are around 1.6°C warmer than the 1961-1990 average conditions over Hans Tausen, and for which the specific SMB of the present-day ice cap is very negative (-0.39, -0.32 and -0.32 m w.e. a$^{-1}$ in MAR3.5.2, RACMO2.3, PDD melt/retention approach respectively). At first the elevation changes at the southern domes are limited
(local SMB is almost at the 1961-1990 level), but as a large mass loss occurs at low elevations the geometry adapts and the rate of ice loss subsequently increases as a result of the SMB – elevation feedback (Figure 11b), until after 1500 years all ice has disappeared. Due to this feedback the volume response time of the ice cap is very short and only amounts to 616 years.





The northern part of the ice cap also changes (see Figure 9e), but the domes remain stable (Figure 11b) and most of the remaining ice mass, corresponding to 19% of the initial mass, is stored here. This is in line with elevation changes derived from ICESat for the period 2003-2008 (Bolch et al., 2013), which indicate that the domes are stable during this time period and even tend to slightly gain mass (typically elevation change up to 0.2 m a$^{-1}$), while the lowest regions are loosing mass at

a high rate (typically more than 0.5 m a$^{-1}$). A more in-depth comparison between these observations and our model results is difficult given our initial steady state assumption and the role of the ice cap response time, but the large-scale features are in reasonable agreement.

For a high emission scenario (IPCC RCP8.5) the 2100 global average surface temperatures are projected to rise by about 4°C

compared to the 1961-1990 average, and over high Arctic regions such as Peary Land temperatures could potentially increase by about +8°C due to the polar amplification (Collins et al., 2013). To simulate the evolution of the ice cap in a warming climate, we consider a +4°C warming, which broadly represents an intermediate emission scenario (in the line of RCP4.5) and a +8°C warming representing a high emission scenario (cf. RCP8.5) (both vs. 1961-1990). For the +4°C scenario and by maintaining the precipitation to the 1961-1990 level, the ice cap entirely disappears within 350-400 years

(i.e. before 2400) (Figure 11a), disregarding whether the forcing is immediately applied at present or linearly until the end of the century. Under the high emission scenario (+8°C) it takes about 140 years for all ice to be gone or 180 years when the forcing is applied linearly, i.e. the last ice disappears in the second half of the 22$^{nd}$ century (Figure 11a). Different model simulations indicate that under such warm conditions the large-scale ice cap evolution is not much affected by its initial state, whether starting from the observed geometry, the 1961-1990 steady state geometry, or a slightly different transient

geometry. Modelling the transient evolution of the ice cap over the last centuries to millennia is therefore of relatively limited interest when it comes to simulating the future mid- to long term ice cap evolution in a (much) warmer climate as the evolution is almost fully driven by the SMB rather than by the ice dynamics.

### 6.2.2 Sensitivity to precipitation changes

Precipitation changes influence the SMB and have the potential to (partly) attenuate the ice loss in the case of warming. In

order to prevent the modelled 20% total mass loss under 1981-2010 conditions, the precipitation has to increase by about 25%: i.e. under these conditions the integrated SMB of the present-day ice cap is again close to 0 m w.e. a$^{-1}$ (see Figure 11a). For 2005-2014 conditions a considerably higher precipitation increase, around 75%, is needed for the present-day 1961-1990 steady state ice cap volume to be maintained, while for the intermediate future warming (+4°C vs. 1961-1990) the precipitation needs a dramatic increase, by about around 340% (Figure 11a). Based on 5 simulations (1961-1990

+0/+1/+2/+3/+4°C) this non-linear relationship is approximated as a 2$^{nd}$ order polynomial (Figure 11c):

$$P = 0.132\ T^2 + 0.316\ T + 1 \tag{11}$$



Where T is the temperature forcing and P the corresponding precipitation forcing (scale factor) (both vs. 1961-1990) needed to prevent a mass loss (vs. 1961-1990 steady state).

### 6.2.3 Implications for future ice cap evolution and geometry

A future increase in precipitation over the ice cap is projected as the surrounding ocean is to become ice-free in a warmer climate and to act as an important moisture source (e.g. Braithwaite, 2005). The 20[th] century conditions are at the borderline between ice-free fjords and fjords with semi-permanent ice-cover (Weidick, 2001) and from palaeo-records it is known that in the warmer mid-Holocene period, when the Arctic ocean was seasonally ice free, the precipitation was up to twice as high compared to present (Madsen and Thorsteinsson, 2001). Based on this the future precipitation increase could be well above the one that would follow from a Clausius-Clapeyron relationship.

It is therefore expected that for a moderately warmer climate, up to 2-2.5°C warmer than the 1961-1990 conditions, the ice loss as a result of a temperature increase may be partly attenuated (cf. Machguth et al., 2013). Under climatic conditions needed to preserve the steady state volume (e.g. 1981-2010 (P+25%) and 2005-2014 (P+75%)) the ice cap total SMB would change only little compared to 1961-1990, but the SMB spatial distribution is more affected. Whereas the temperature increase mainly decreases the SMB in the present-day lower areas, the temperature increase on the higher areas leads to higher precipitation and a higher SMB. As a result the steady state ice cap margin retreats (smaller steady state area), whilst the interior thickens, resulting in a steeper ice cap (Figure 11d). This is in agreement with recent ICESat observations on Arctic ice caps, which indicate a marginal ice loss and local thickening (for the interior), as is the case for instance for the Austfonna ice cap (Svalbard) (Moholdt et al., 2010), the Flade Isblink ice cap (Greenland) (Rinne et al., 2011; Bolch et al., 2013) and also the Hans Tausen ice cap (Bolch et al., 2013). An in-depth comparison between our modelling study and these observations is again not possible given the differences in timing and the model setup, but our simulations show the potential to reproduce the observed trends and the implications this can potentially have on the future ice cap evolution.

For even warmer conditions (>3°C) the required precipitation increase of more than 200% needed to counteract the mass loss is much higher than expected from climate modelling and palaeoclimatic records and as a consequence the ice cap will in all cases (largely) disappear. This is for instance clear from the 1961-1990 (+4°C, P+100%) simulation, where only very little ice survives, corresponding to around 2.5-3% of the present-day volume (see Figure 11a). For the high emission scenario (+8°C vs. 1961-1990), all ice disappears, and even under an extreme precipitation increase of for instance 200% it would only take 10-15 years longer for all ice to be gone compared to a constant precipitation scenario.





### 7 Conclusions and recommendations

In this study a SMB – thermomechanical ice flow model was developed for Hans Tausen Iskappe, the world's northernmost ice cap, and tested for various parameter settings and model complexities. Despite the remoteness of the ice cap a large dataset was available encompassing ice thickness (at high spatial resolution, not included in the Greenland datasets), surface

mass balance measurements (and related temperature and precipitation measurements), ice temperature measurements and surface velocities (both from the field and remote sensing techniques). The numerical simulations were tuned and validated based on this data set and provide us with valuable insights in the dynamics of the ice cap. Our main findings and their implications for the dynamics and modelling of other Arctic ice caps are:

1) RCM RACMO2.3 precipitation agrees well with field observations and so does the reconstructed SMB, which is a valuable contribution to the model validation as much of the northernmost parts of Greenland have little observations and mass balance measurements. With our simple PDD melt-retention approach, using downscaled RACMO2.3 precipitation, we were able to reproduce the measured SMB and the modelled SMB in good agreement with output from other RCMs, and this for different periods in time. The simple PDD melt-retention model allows for a direct coupling of the ice topography and

the SMB, which is crucial for the ice cap dynamics.

2) For solving the ice dynamics and the ice flow in the fast-flowing outlet glaciers a higher-order solution is needed. Compared to a local solution (SIA), this influences the steady state volume in the order of 6-8%, and also the area is affected around 2-3% through the SMB – elevation feedback. We also show that to reproduce the observed surface velocity patterns

in outlet glaciers a higher-order solution is needed as the contrast between the high and low velocities is overestimated with a local solution (SIA). When modelling ice caps with fast outlet glaciers a non-local velocity solution may be worthwhile depending on the focus of the study and the availability of high-resolution field data. For understanding the large-scale dynamics of ice caps and their long-term evolution the focus should rather be on an accurate representation of the SMB than on the ice dynamics and under most cases a SIA is justified. The effect of running the model at a higher spatial resolution

(250 m vs. 500 m) also mainly affects the outlet glaciers but was found to be overall rather limited.

3) Under the 1961-1990 climatic conditions the ice cap evolves to a steady state that is close to observations in terms of ice cap geometry (extent and ice thickness), ice temperature and surface velocities. This is in agreement with output from RCMs and field observations that indicate that little changes occurred during this period. Given the long response time of the ice

cap a statement about its equilibrium with the 1961-1990 climatic conditions cannot be made, but likely the limited changes result from an interplay between a long-term growth trend, linked to Holocene cooling, and a short-term retreat trend, from the end of the LIA, linked to a warming.





4) Both englacial temperature measurements, modelled ice thickness and temperatures in outlet glaciers suggest that there is an important heating mechanism related to infiltrating meltwater in the ablation area of the ice cap. Without this additional heating source the measured temperatures in the outlet glaciers cannot be reproduced and the ice thickness is locally strongly overestimated. In this study we provide additional evidence related to extensive warming through meltwater, a mechanism that could be of large importance when modelling the dynamics of Arctic ice caps, especially in a warming climate, with more surface melt and a potential higher meltwater supply to the base.

5) The SMB-elevation feedback is a crucial mechanism for the ice cap evolution and stability. Due to this feedback the southern part of the ice cap is extremely sensitive to a change in climatic conditions. This is clear from its total disappearance when the 1961-1990 climatic conditions are warmed by more than 0.85°C. This is in line with palaeorecords that suggest that the southern part of the ice cap totally disappeared during the Holocene Thermal Maximum. The northern part of the ice cap is situated on a higher plateau and the local ice thickness is lower and this part is therefore less affected by the SMB-elevation feedback and more stable. The SMB-elevation feedback is also responsible for thresholds in the system under certain conditions, where the final steady state depends on the initial geometry, and for which the ice flow from the northern plateaus to the southern part of the ice cap is a crucial factor. This ice flux also causes the response time of the ice cap to be up to several thousands of years under some particular conditions. For cases where this ice feeder-supplier mechanism is more limited the response time is typically around 1000 years, although this can be up to a factor 2 smaller under the influence of the SMB-elevation feedback. These time scales are in agreement with palaeorecords that suggest that the ice cap (largely) disappeared during the Holocene Thermal Maximum and subsequently started to regrow some 3500-4000 years ago.

6) For limited SMB perturbations the ice cap evolves to a steady state and does not have a run-away behaviour as is occurring in some ice cap modelling studies, where the ice cap has a tendency to grow far beyond the observed state or evolves to a very small ice cap for the slightest perturbations. This is in part related to the specific geometric setting, where the northern part, situated on a higher plateau, delivers its mass surplus to the lower-lying southern part. On the other hand, the fact that the SMB is modelled explicitly ensures that for the highest parts of the ice cap the SMB only changes little under different climatic conditions (e.g. slightly increases under colder conditions), as is the case in reality, avoiding artefacts inherent to simple parameterizations of the elevation dependence of SMB.

7) In a moderately warming climate (up to 2-2.5°C vs. 1961-1990) the projected mass loss may be partly attenuated if precipitation sharply increases. A local ice core drilled at Central Dome suggests that precipitation was higher during the Holocene Thermal Maximum, and this will likely also be the case in a warmer climate, with more ice-free ocean conditions. Due to their high elevation the local domes are almost unaffected by a moderate temperature rise and as a consequence of a precipitation increase they could gain mass, making the ice cap steeper, which is in line with recent satellite observations.

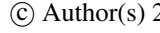



For warmer conditions (>3°C warming) the ice cap will (almost) fully disappear, even under a higher precipitation regime, within 350-400 years (1961-1990 +4°C) to within less than 200 years (1961-1990 +8°C). This evolution is almost irrespective of the modelled initial conditions. Taking into account the inherent uncertainty of the SMB model, there is no need for a late Holocene transient run and detailed future scenarios for understanding the potential future ice cap evolution

and the potential for the precipitation to mitigate this.

**Author contribution**

H. Zekollari and P. Huybrechts designed the experiments and H. Zekollari performed the fully-coupled model simulations. B. Noël, W.J. van de Berg and M. Van den Broeke provided the RACMO3.2 model simulations and developed the downscaling methods for this output. H. Zekollari and P. Huybrechts wrote the manuscript. B. Noël, W.J. van de Berg and

M. Van den Broeke read the manuscript and provided valuable comments.

**Acknowledgments**

We thank A.M. Solgaard, A.P. Ahlstrøm and C. Hvidberg for their help in retrieving all data from fieldwork and for providing documentation from the expedition in the 1990s. H. Zekollari wants to thank H. Goelzer for his generous support with technical issues. H. Zekollari holds a PhD fellowship of the Research Foundation – Flanders (FWO-Vlaanderen).

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



**Tables**

| Location (and elevation in m a.s.l.) | 1975-1995 average annual accumulation (from shallow ice core) (m w.e. a$^{-1}$) | 1975-1995 average annual precipitation from RACMO2.3 (m w.e. a$^{-1}$) |
|---|---|---|
| North Dome (1318 m) | 0.27 | 0.24 |
| Central Dome (1275 m) | 0.09 | 0.11 |
| BH75 (1150 m) | 0.11 | 0.13 |
| BH76 (1125 m) | 0.10 | 0.15 |

**Table 1**. Comparison of measured accumulation and modelled precipitation (RACMO2.3, 1-km resolution) for the period 1975-1995. Site location is shown in Figure 1.



**Figures**

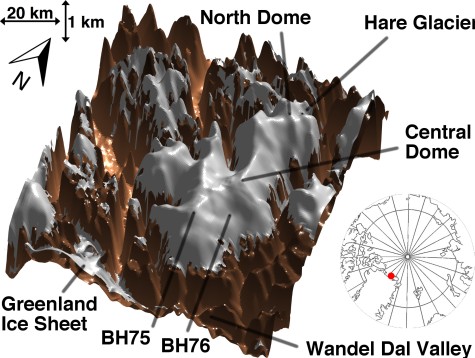

**Figure 1.** Hans Tausen Iskappe in the mid 90s. Figure created with the TopoZeko toolbox (Zekollari, 2016). Map in lower right corner

5    shows the location (red dot) of the ice cap in Greenland.

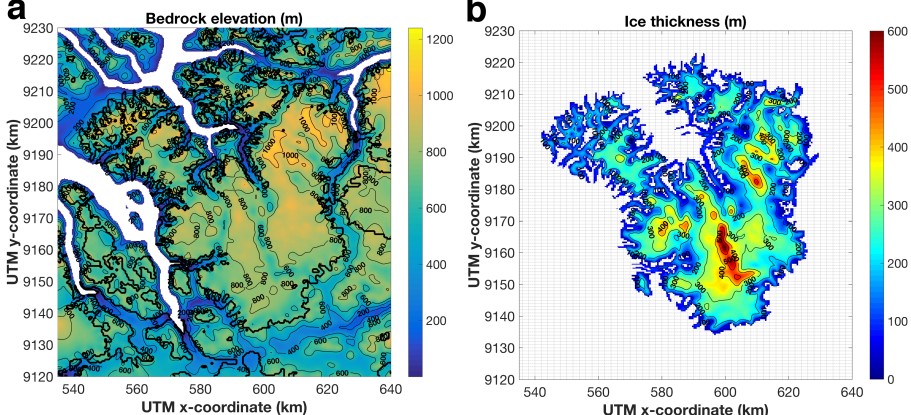

10   **Figure 2.** (a) Bedrock elevation (Starzer and Reeh, 2001). Areas below sea level (fjords with semi permanent sea-ice) are depicted in
     white, the thick black line corresponds to the outline of the observed glaciated area. (b) Ice thickness in the mid 1990s based on the DEM
     from Starzer and Reeh (2001).



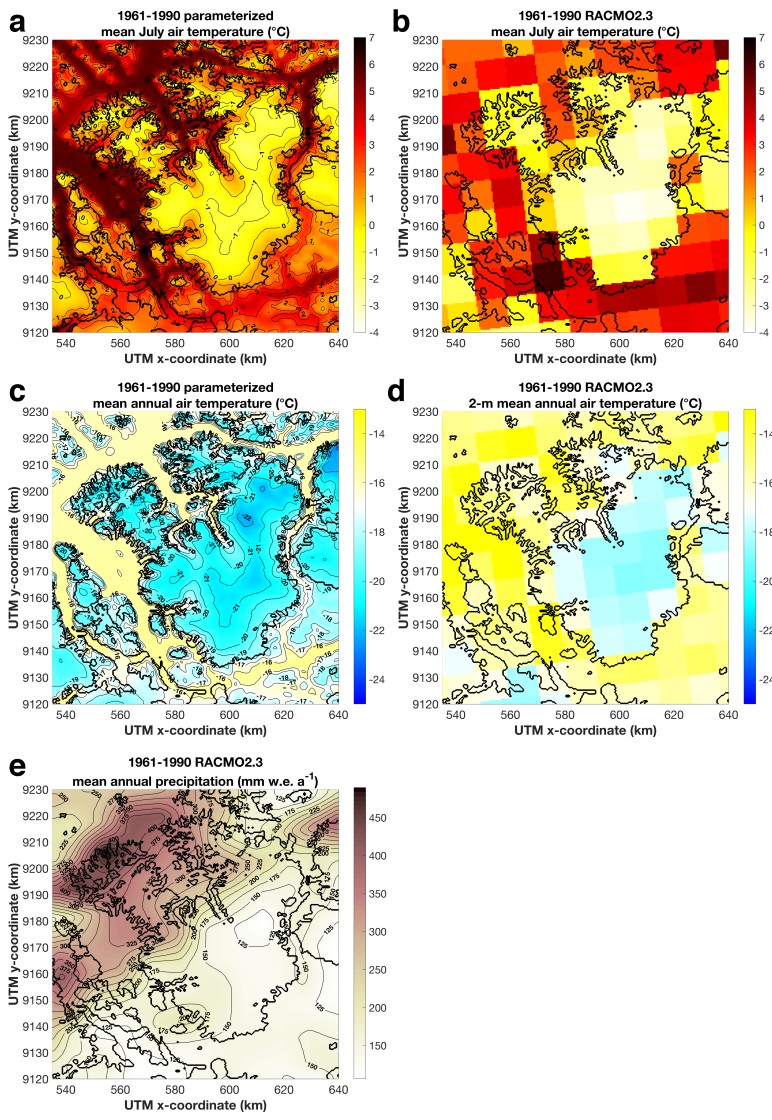

**Figure 3.** (a) 1961-1990 parameterized mean annual temperature, (b) 1961-1990 RACMO2.3 mean annual temperature (11 km resolution), (c) 1961-1990 parameterized mean July temperature, 1961-1990 RACMO2.3 mean July temperature (11 km resolution) and (e) 1961-1990 RACMO2.3 mean annual precipitation (1 km resolution). In all figures the thick black line corresponds to the outline of the observed glaciated area (Starzer and Reeh, 2001).



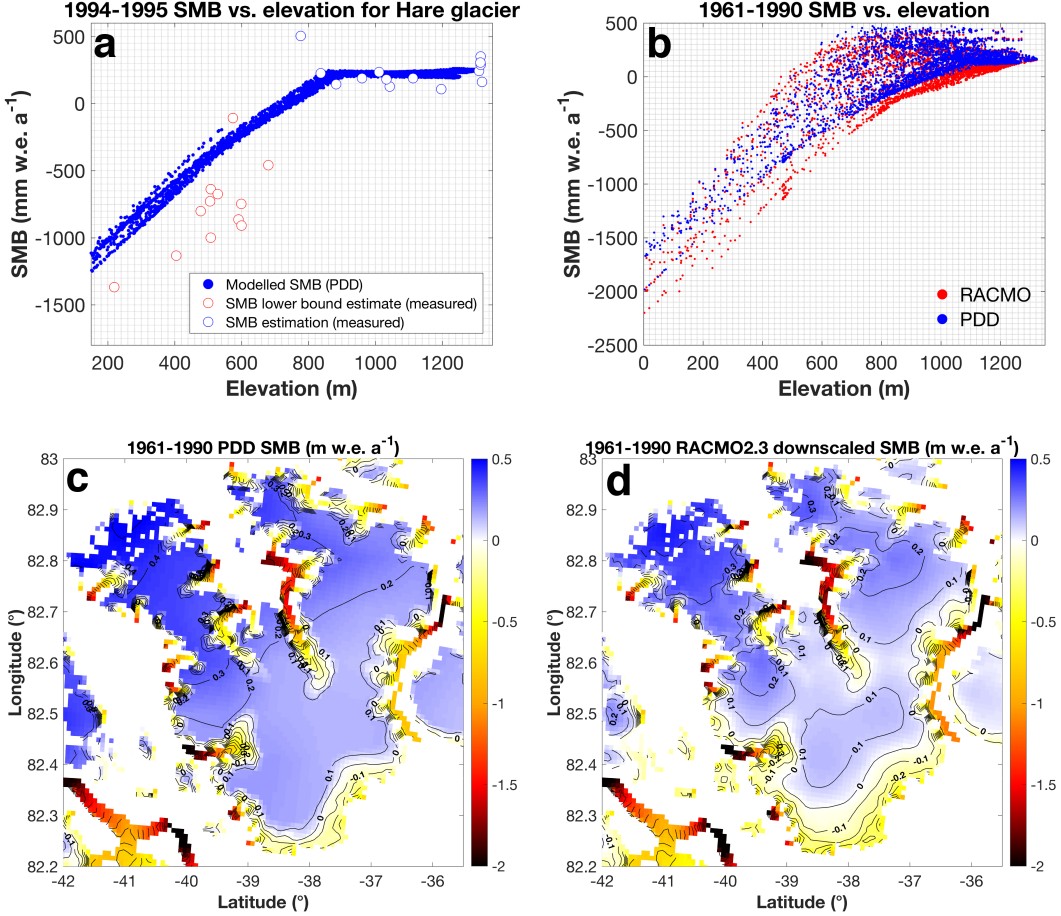

**Figure 4.** (a) 1994-95 SMB for Hare glacier based on the PDD melt/runoff model and SMB measurements (lower bound estimate in the ablation area, see text). (b) SMB versus elevation for the period 1961-1990 for the PDD melt/runoff model and RACMO 2.3. Average SMB for the period 1961-1990 from (c) PDD melt/runoff model and (d) RACMO 2.3 RCM model. The masking and the calculations for figure (b,c,d) are based on the 1 km GIMP DEM ice mask and topography (Howat et al., 2014).



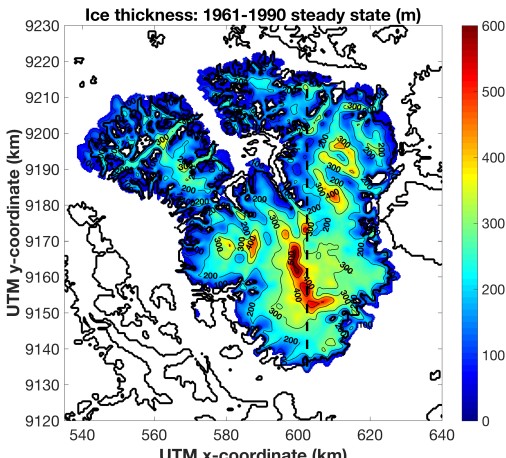

**Figure 5.** 1961-1990 steady state ice thickness from the HO 250-m resolution run. The thick black lines represent the outlines from the glaciated areas from the DEM (Starzer and Reeh, 2001). The dotted black line is the transect at X UTM = 602.5 km that is illustrated in

5    **Figure 11**.

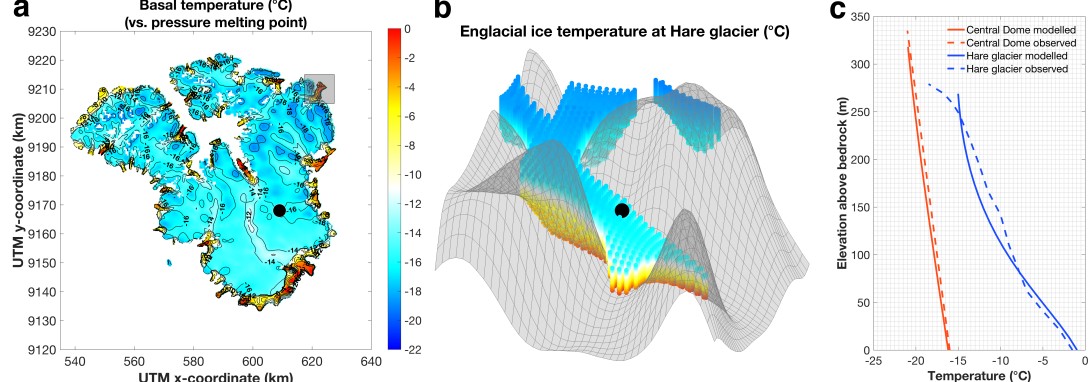

**Figure 6.** (a) Modelled basal temperatures for the 1961-1990 steady state ice cap. The shaded grey box represents Hare glacier, the area shown in (b), the black dot represent the drill site at the Central Dome. (b) Englacial temperatures at Hare glacier, the color scale is the

10    same as in (a). The black dot represent the location of the drill site where englacial temperatures were measured. (c) Modelled and measured (observed) temperature profiles for the Central Dome and at Hare glacier.



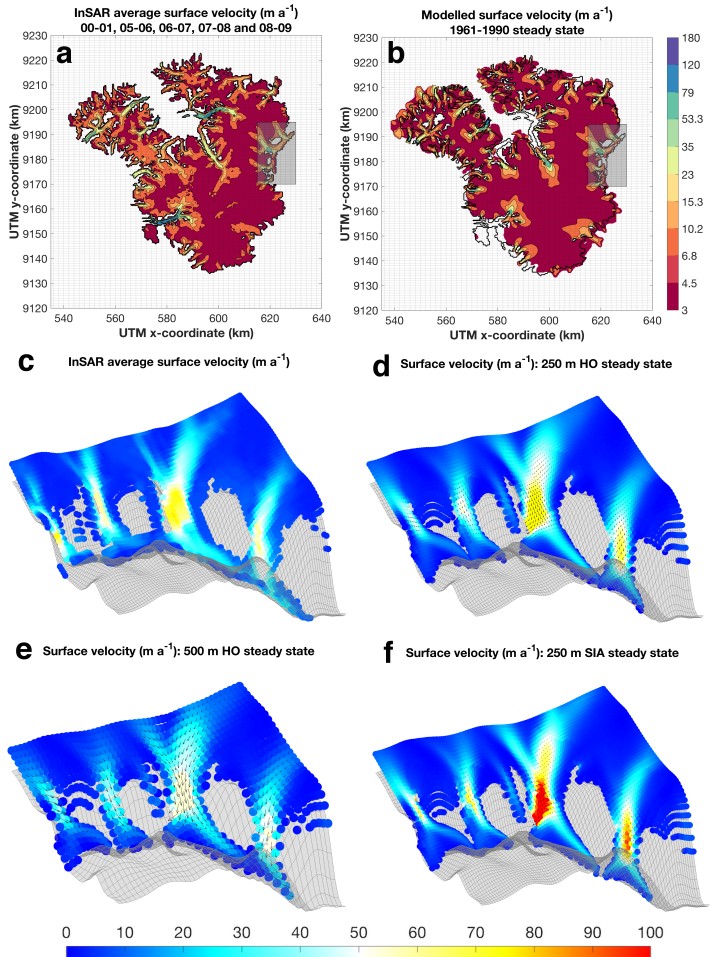

**Figure 7.** (a) InSAR derived surface velocities (Joughin et al., 2010, 2015) and (b) 1961-1990 steady state surface velocities (250-m HO run). Black lines represent the observed ice cap outline from the Starzer and Reeh (2001) DEM, the shaded box delineates the area shown in (c,d,e,f). (c) InSAR derived surface velocities, (d) 250 m HO surface velocities, (e) 500 m HO surface velocities and (f) 250 m SIA surface velocities. For the InSAR velocities (c) the geometry corresponds to the observed one (Starzer and Reeh, 2001), while for the model runs (d,e,f) the geometry corresponds to the steady state geometry.



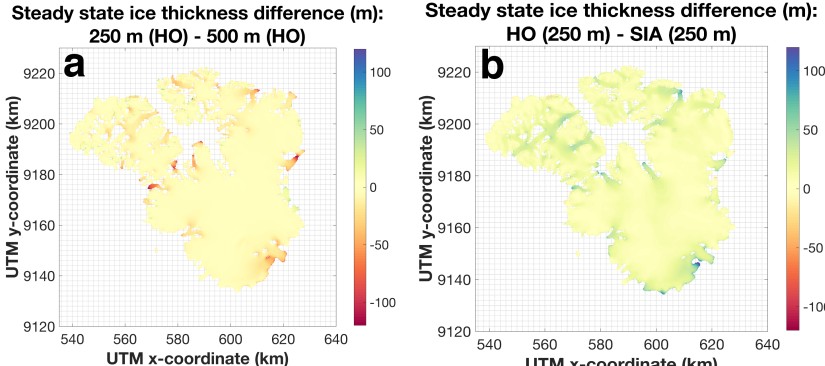

**Figure 8.** (a) Difference in ice thickness between 250 m and 500 m HO steady states and (b) difference in ice thickness between HO and SIA steady states (at 250 m).

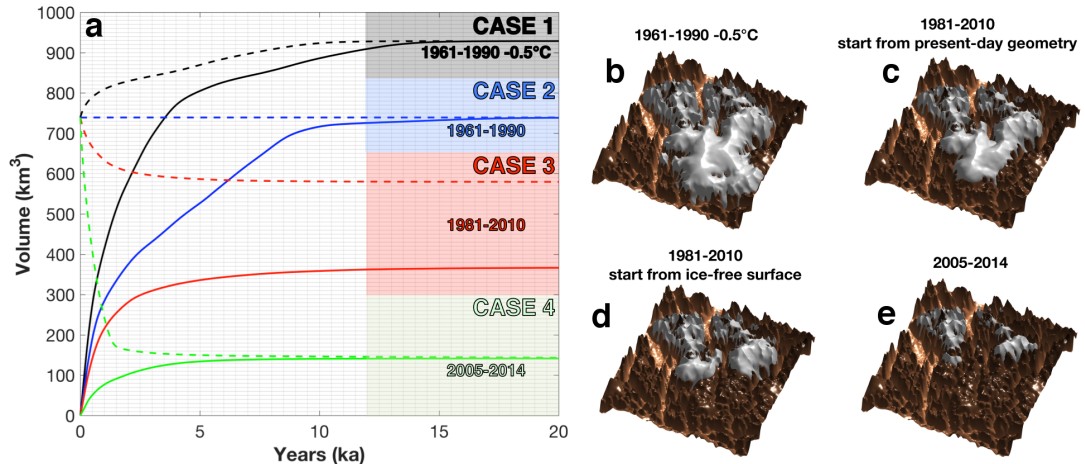

**Figure 9.** (a) Volume build-up of Hans Tausen Iskappe for different initial states (ice free surface and 1961-1990 steady state geometry) and under different climatic conditions. The four coloured regions (cases) represent clusters of simulations with a similar build-up and thresholds in the system and are defined and discussed in the text. (b,c,d,e) Modelled steady state geometry for different climatic forcings (figures made with TopoZeko toolbox (Zekollari, 2016))





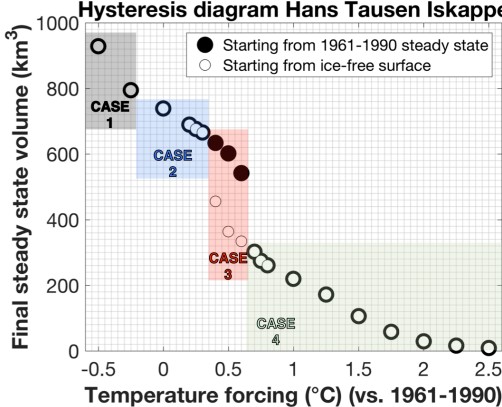

**Figure 10.** Final steady state volumes under different climatic conditions and for different starting geometries. The four cases are the same

5   as in **Figure 9** and are described in the text. Bold circle outlines mean that the final steady state is independent from the initial conditions.



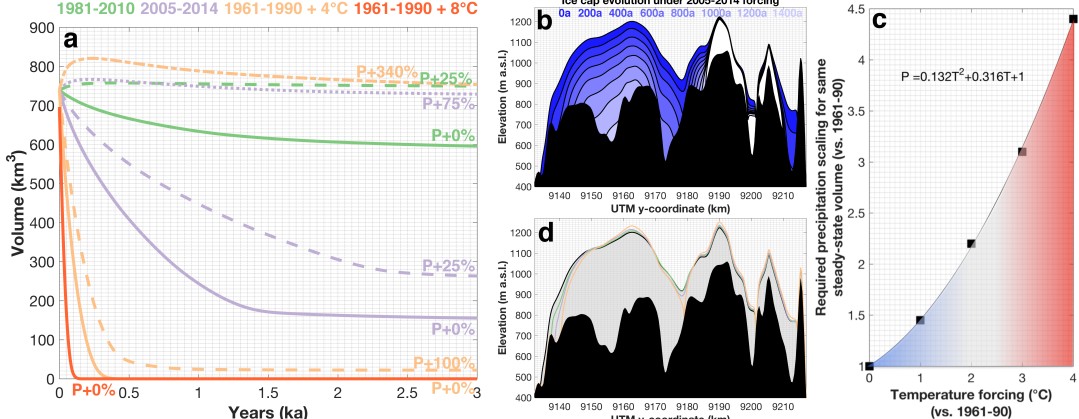

**Figure 11.** (a) Volume evolution of Hans Tausen Iskappe under different climatic conditions. The solid lines represent simulations where

5 the precipitation is unaltered (i.e. P+0%), while the dotted lines represent simulation with a modified precipitation. The starting point of all simulations is the 1961-1990 steady state geometry. (b) Evolution of the Hans Tausen Iskappe along UTM x= 602.5 km transect under the 2005-2014 forcing, starting from the 1961-1990 steady state, for 200 year time intervals. The black area represents the bedrock and the white area represent the final steady state geometry. The location of the transect is shown in **Figure 5**. (c) Temperature forcing and corresponding precipitation forcing (scale factor) needed for 1961-1990 steady state volume to be preserved. The polynomial fit is based

10 on 5 simulations (1961-1990 +0/1/2/3/4°C), which are represented by the black dots. The blue area broadly corresponds to the range where an attenuation of the mass loss is possible, whilst the red area represent the range under which the ice cap is to (largely) disappear. (d) Ice cap profiles along the UTM x = 602.5 km transect. The shaded grey area is the 1961-1990 steady state ice cap geometry. The 3 other geometries correspond to the 1981-2010 (P+25%), 2005-2014 (P+75%) and 1961-1990 +4°C (P+340%) steady states and follow the same colour scheme as in (a). The location of the transect is shown in **Figure 5**.