# Peer review of "Sensitivity, stability and future evolution of the world's northernmost ice cap, Hans Tausen Iskappe (Greenland)"

_The Cryosphere, 2016_

## Referee Comment (RC1) · Anonymous Referee #1 · 12 Jan 2017

In their paper *Sensitivity, stability and future evolution of the world's northernmost ice cap, Hans Tausen Iskappe (Greenland)*, Zekollari et al. present results from a suite of high-resolution higher-order ice sheet model simulations. I very much enjoyed reading this manuscript as it describes a set of well-designed experiments and is well-written. I am confident that with a moderate amount of editing this publication will be a valuable contribution to the field of arctic glaciology.

General comments:

Several of the figures need a bit of work to make them easier to read. I will provide specific comments below.

Page 6, lines 6 and following: The description of where positive SMBs are permitted is somewhat unclear. (Do I get it right, that a positive SMB on an ice-free area is only permitted, where there is present-day ice cover?) Please rephrase.

Please spend a few lines on why you decided not to take the RACMO temperatures in combination with a lapse rate, but your analytical expression.

Please spend a few lines on how the bedrock elevations were obtained / interpolated. This is one of the key fields for ice flow modeling and low data quality in certain areas might explain velocity mismatches (c.f. Aschwanden et al. (2016)). I expected a few words on this in the model setup section. Can you specify how highly resolved they are (in terms of smallest features that are/can be resolved, not in terms grid spacing in the file)?

While you write that there are several shallow cores from which a precipitation parametrization was derived, you only compare the RACMO data to four cores in Table 1. Are these all cores that can be compared? If not, why/how were they selected? For more data, a scatter plot could be useful.

In section 4.2 please provide the MAR SMB for comparison. Otherwise the main message is a sign flip. There is some overlap with section 5.1.1. Please clean this up.

Section 5.1.4: Disagreement on the ice thickness / surface elevation might not just be the cause of a velocity error. It might also be a consequence...

I don't fully understand why you kept the SMB constant in the 500 m grid resolution experiment. Am I correct in assuming that the SIA experiment was performed with SMB-elevation feedback?

Section 6.2.1
Can you provide summer temperature changes for this region from the CMIP5 RCP8.5 simulations? Is there a matching RACMO-experiment? Polar amplification usually is strongest in winter, which is of little significance to the ice SMB.

Figure 5:

Please (also) plot the difference between your modeled and the observed ice cap thickness.

Technical comments:

**Page 1**

Line 14:
Please flip the direction of the comparison (SIA is the *erroneous* experiment) and then replace *modifies* with *decreases* (if I got the direction right in the main part of the manuscript). A new text would then be something like *Using the Shallow-Ice Approximation decreases ...*. I would actually prefer omitting the entire sentence, as I don't see it as relevant to your main message.

Line 22:
Please replace *corresponding* with a word that clearly describes causality (*following?*).

Line 26
I think, it should read *disappear around 2400 and 2200 A.D.* **respectively**,
Replace *irrespective* with *independent* (also in other locations in the text).

**Page 3**
Line 29:
*often terminate up to several hundred meters* is vague in multiple ways. Also, often is with respect to time, not the individual glacier. Maybe replace with *many of them terminate several hundred meters ...*?

Eqn (5)
The left hand side should read $\partial_z \tau_{iz}$ ($\partial_z$ instead of $\partial_i$), e.g. eqn 5.76 of Greve and Blatter (2009)

**Page 6**
Line 19:
*sub/* should probably by *sub-*.

Line 20:
Please specify more details on the firn warming. Are we talking about firn modeled by your PDD model?

**Page 9**
Line 21:
please convert i.e. to w.e. (even if it means keeping the number unchanged).

Line 22:
Please change *occurred* to *were performed* or something similar.

**Page 14**
Line 2:
Importance **of** initial conditions

Lines 20/21:
Please rephrase *the shape of the volume evolution curve is far less exponential compared to case 1*. Maybe *the growth is slower than in case 1*?

**Page 15**
Line 11:
Remove *largely*. Do you mean *strongly*? Maybe provide the area change in %?

**Page 16**
Line 30:
I would suggest using $\Delta T$ instead of $T$.

**Page 17**
Line 21: I would suggest pulling the reference to Bolch et al. (2013) to the front of the list, as it is the ice cap under investigation in this manuscript.

**Figures**

Several of the figures have a very dense raster in the background. I find it way more disturbing than helpful. Please reconsider.

**Figure 1**
I found the figure hard to read. Maybe you could decrease the vertical exaggeration?

**Figure 3**
(a and b) maybe a discrete colormap would make the comparison between a and b easier. This way, it is virtually impossible to read temperature values from Fig 3b. Maybe you could display RACMO downscaled with the same lapse rate as used in the analytical expression (and with bilinear/... interpolation between the grid cells)? This would make the comparison of the two plots a lot easier. Same for (c,d). Both temperature plots could (should) use the same colormap. The colormap in (c,d) very prominently marks the difference between regions with annual mean temperatures above or below -15C. Is there a specific reason for this, otherwise a linear colormap might be better.

**Figure 6b**
What are you telling the reader with Fig. 6b? I think, it can be omitted without loss of information.

**Figure 7**
In 7 a and b you could zoom in a bit more on the ice covered domain. Then it would be displayed bigger and easier to read. I would suggest using the same color scale for the top and bottom figures. Currently one uses linear and the other one log scaling, although the data ranges seem pretty much identical. The color scale in Fig 7 c-f has large ranges where the color hardly changes at all and then sharp transitions (looks like I'd have difficulty telling 60 from 75 m/yr, but 75 to 85 is very clear). A more linear color scale (or focus on the low velocities as in a log scale) would be more helpful.

The 3d-projection in c-f does not seem to add information. To me it's rather confusing.

Most likely a 2d plot would do a better job.

**Figure 8**
Please add "resolution" in the caption to make clear that 250 m and 500 m refer to the model resolution.

**Figure 9**
Again, I find the 3d-Plots very hard to read. They appear very dark. I don't think it's necessary to cite yourself again for the plot tool. The reference in Fig. 1 should be enough.

**Figure 11**
I find Fig. 11a extremely hard to read as there is minimal contrast between the colors of lines and text and the background raster. Please change this. The same color problem applies to Fig. 11d. What is the color shading in Fig. 11c about? This should be a line plot.

Greve, R., & Blatter, H. (2009). Dynamics of Ice Sheets and Glaciers. Dordrecht [u.a.]: Springer.

---

## Referee Comment (RC2) · Anonymous Referee #2 · 2 Feb 2017

General comments

This paper describes a modelling study for an ice cap in northern Greenland, outside the main ice sheet. It applies a coupled ice flow – mass balance model with PDD – runoff/retention model and precipitation downscaled from RACMO. The experiments are well designed and conclusions drawn from this study are interesting, one being that there is no gain in performing time dependent simulations for the ice cap, Holocene or future, as the response of the model is independent of the model initial conditions and there appears to be 4 sets of steady states possible for the ice cap. The model study shows hysteresis response for a narrow band of temperature forcing (around 0.5°C higher temperature than 1961-1990 average). Authors also point out that the SMB-

elevation feedback is a crucial mechanism for the ice cap evolution and stability and that infiltrating meltwater in the ablation area is necessary in order to simulate englacial temperatures. The paper is clearly written and the conclusions are clear, some minor comments for improvements are suggested below. To improve the overall clarity of the paper, better description of the model resolution as well as the temperature forcing would be good, see below.

Specific comments:

It is now clear whether the input fields for the PDD scheme, precipitation from RACMO and parameterized temperatures are downscaled to the ice flow model grid resolution or if the SMB field is downscaled. Clarification in the model setup section is needed, see suggestions below. In several places it is stated that RACMO is run at 11 km and then precipitation downscaled to 1 km, but is it then further downscaled to 500 m or 250 m?

Some confusion is in the discussion of the results, the description of the forcing is not clear, are all the forcing scenarios shifted relative to the 1961-1990 condition? And then some of them fall onto 1981-2010 or 2005-2014 mean conditions? Figure 9 has both 1961-1990—0.5°C and different periods, but Figure 10 has all temperature scenarios relative to the reference. Some clarification in the model or experimental description is needed.

Technical corrections:

Page 1, Abstract, Line 24, suggest to replace "grow" with "thicken"

Page 2, line 5, delete "s" on exist

Page 3, line 15-17, sentence is confusing, how do inconsistencies arise with bedrock from Starzer and Reeh (2001) if the direct ice thickness measurements are not included in the Bamber et al (1013) dataset?

Page 6, line 31-32, make sure that the minus sign - sticks with the number Page 7, line

24, suggest to add "daily" before variability

Page 8, here the resolution of the T parameterization could be mentioned, ice flow model resolution?

Page 9, line 11, is the precipitation then further downscaled to ice flow model resolution? Page 9, line 20, what is "frontal SMB" - terminus ablation? Page 9, line 27-32 – here the different forcing scenarios, shifted relative to 1961-1990 or other periods, could be presented and explained

Page 10 line 6, suggest to replace "further" with "below" Page 10, line 12, what does "imposed" mean here? is the SMB regridded to 250 or 500 m resolution? Page 10, line 21, edit, something strange in the sentence "but except" Page 10, line 31, is the grid not at 250m resolution?

Page 11, line 10, suggest to replace "heating" with "heat flux" Page 11, line 21 and onward, it is not clear what is discussed here, paragraph needs rewriting "With a value of 150 mW m-2 . . .." - is not clear

Page 12, lines 1-4, strange sentence, suggest editing Page 12, line 17, suggest to replace "to" with "with" Page 12, line 17-20, too long sentence, suggest to split into two

Page 13, line 10, here is seems like SMB is calculated at the ice flow model resolution, is then T and P downscaled to this resolution? Page 13, lines 18-21, long sentence, suggest to split or rewrite Page 13, line 25, "which results in smoothing. . ."

Page 14, line 2, something is missing in title, "of"? Page 14, line 23, text is confusing "(vs. " do you mean relative? Is it only temperature shift?, what are 1981-2010 conditions, do you mean in the RACMO model? Page 14, line 28, "build up faster than under 1961-1990 condition" – does not make sense, what do you mean by faster?

Page 15, line 5, replace "parts" with "part" Page 15, line 15, what does "largely similar" mean?

Page 16, line 15, needs editing, text is unclear Page 16, line 19 "slightly different transient geometry" needs editing, not clear

Page 17, line 20-22, text needs editing, two times "potential" in sentence

Page 18, line 4, suggest to replace "was" with "is" Page 19, line 1 "Both" – and then three things are mentioned, needs editing

Figure 3 Annual and mean July are in different sequence in the figure and in the caption; (a), b) vs c) d)), suggest to move the precipitation figure to the right column so that RACMO fields are all in the same column

Figure 5, suggest coloring dotted line white, it is hardly visible as it is now

Figure 6 b) is not clear, suggest less vertical exaggeration?

Figure 9 C is the period 1961-1990? Or 2010?

Please also note the supplement to this comment:
http://www.the-cryosphere-discuss.net/tc-2016-271/tc-2016-271-RC2-supplement.pdf

---

## Author Comment (AC1) · 28 Feb 2017

In their paper *Sensitivity, stability and future evolution of the world's northernmost ice cap, Hans Tausen Iskappe (Greenland)*, Zekollari et al. present results from a suite of high-resolution higher-order ice sheet model simulations. I very much enjoyed reading this manuscript as it describes a set of well-designed experiments and is well-written. I am confident that with a moderate amount of editing this publication will be a valuable contribution to the field of arctic glaciology.

We thank the reviewer for his positive general appreciation of the manuscript.

**General comments:**

Several of the figures need a bit of work to make them easier to read. I will provide specific comments below.

The comments regarding the figures have all been addressed and the figures were updated accordingly.

**Page 6, lines 6 and following**: The description of where positive SMBs are permitted is somewhat unclear. (Do I get it right, that a positive SMB on an ice-free area is only permitted, where there is present-day ice cover?) Please rephrase.

Basically there are three regions:
  i. Regions covered by the present-day ice cap: here the ice cap can freely grow
  ii. Ice-free regions within the present-day ice cap: here no ice can build up and this is imposed (in order to represent the removal of accumulation through processes that cannot be caught by our model, such as wind drift)
  iii. Regions outside the present-day ice cap. Here the ice cap can grow (and to answer the reviewer's comment: the surface mass balance can also be positive in this case!), as long as the particular grid cell is connected to Hans Tausen Iskappe and the bedrock elevation is above -50 m. i.e. the ice cap can only expand from its interior, and not grow from neighbouring ice masses (as these cannot be modelled as they are not entirely part of the domain).

In order to emphasize the latter point (iii) an additional passage was added:

*The ice can subsequently expand freely, without any constraints (e.g. can connect to the GrIS), and both negative and positive surface mass balance can thus be obtained for areas outside the present-day ice cap. The ice cap cannot expand for areas where the bedrock elevation is lower than -50 m, where the ice is removed to crudely represent calving.*

Please spend a few lines on why you decided not to take the RACMO temperatures in combination with a lapse rate, but your analytical expression.

The main reason for not taking the RACMO temperatures, but rather opting for the analytical expression is twofold:
  i. RACMO temperatures contain an imprint of the present-day ice cap geometry and surrounding ice masses. This imprint should not be present for the long timescales considered in this study (up to several thousands of years), having very different geometric settings.
  ii. The analytical solution is flexible in its application and can directly be

applied at any model resolution without a need to rely on a downscaling (which would be the case if the RACMO data was to be used)

This is now also explained in the updated manuscript:

*A temperature parameterization is preferred over lapse-rate corrected RCM temperatures to remove the bias from the present-day imprint of the ice cap on its own temperature field in a different geometric setting. Furthermore the temperature parameterization is flexible and allows for a direct application at different resolutions, without the need for complex downscaling methods (needed for RCM data)*

However note that given the similarities between the RACMO and the parameterized temperatures (see Figure 3 and section 4.1.2.), a relatively similar SMB would have been obtained if the RACMO temperatures had been used in the PDD model, using the same lapse rate).
* * *
Please spend a few lines on how the bedrock elevations were obtained / interpolated. This is one of the key fields for ice flow modeling and low data quality in certain areas might explain velocity mismatches (c.f. Aschwanden et al. (2016)). I expected a few words on this in the model setup section. Can you specify how highly resolved they are (in terms of smallest features that are/can be resolved, not in terms grid spacing in the file)?

This is valuable point raised by the reviewer. The bedrock DEM was constructed by Starzer and Reeh (2001) by a direct interpolation for the interior (with a dense direct measurement network) and a parameterization for the outlet glaciers (where measurements are scarcer). This is now mentioned in the updated manuscript:

*Whereas the interior of the ice cap has a dense network of ice thickness measurements (up to several points per square kilometer), measurements on the outlet glaciers are scarcer and here a parameterization relating the surface slope to the ice thickness is used (Starzer and Reeh, 2001)*

Given the limited number of direct measurements in the outlet glaciers, a part of the model-observation discrepancy may therefore be related to errors in the bedrock DEM. We also mention this at the end of section 5.1.3 (where the observed and modelled ice cap geometry are compared):

*Notice that given the limited amount of direct ice thickness measurements in the outlet glaciers (Starzer and Reeh, 2001), part of the model-observation discrepancy may be related to local errors in the bedrock DEM.*
* * *
While you write that there are several shallow cores from which a precipitation parametrization was derived, you only compare the RACMO data to four cores in Table 1. Are these all cores that can be compared? If not, why/how were they selected? For more data, a scatter plot could be useful.

In the original manuscript this was indeed not clearly defined. There are only four cores available that span over several decades. To clarify this in the updated manuscript we do not refer to 'several' ice cores anymore, but to 'four' ice cores:

*The accumulation has been derived from field measurements and four shallow cores that cover most of the 20th century (Reeh et al., 2001)*

**In section 4.2** please provide the MAR SMB for comparison. Otherwise the main message is a sign flip. There is some overlap with section 5.1.1. Please clean this up.

Good point, as the sign flip is indeed not our message at all! The MAR SMB is now also mentioned in section 4.2:

*In another widely used RCM for Greenland, MAR3.5.2 (20 km run, downscaled to the 5 km Bamber et al. (2013) DEM) (Fettweis et al., 2012) an integrated SMB of +0.03 m w.e. a$^{-1}$ is obtained. Given the different topographic input a direct comparison between with RACMO2.3 and the PDD approach is difficult, but also here the RCM output suggests a near-zero SMB for this period.*

Section 5.1.1 was cleaned up to avoid any overlap:

*The modelled limited areal changes under the 1961-1990 average conditions are supported by the RCM output that indicates a near-zero average integrated SMB over this period (see section 4.2)*

**Section 5.1.4**: Disagreement on the ice thickness / surface elevation might not just be the cause of a velocity error. It might also be a consequence...

This is true and is in fact a 'chicken-and-egg' problem. We now also mention this:

*Notice that as the surface velocities and the modelled geometry are related, the surface velocity discrepancy may be a consequence of the geometry discrepancy. The inverse may however also be true: i.e. the surface velocity discrepancy is the cause for the geometry discrepancy.*

I don't fully understand why you kept the SMB constant in the 500 m grid resolution experiment. Am I correct in assuming that the SIA experiment was performed with SMB-elevation feedback?

In both cases (for the 250m/500m comparison and the SIA/HO comparison) the SMB is kept constant to produce the figures (7d,e,f and 8). In this way a 'clean' visual comparison is possible. This is indeed somewhat confusing, as later in the text the effect of the SMB-elevation feedback is discussed (when comparing the area and volume). In order to make this clearer, the reference to the constant SMB is removed from the text (where it is not used), and it is only included in the captions of figure 7 and 8:

*Notice that the SMB is fixed in time (1961-1990 climatology applied on the present-day geometry) in order to make a 'clean' comparison and avoid effects related to the SMB-elevation feedback.*

**Section 6.2.1** Can you provide summer temperature changes for this region from the CMIP5 RCP8.5 simulations? Is there a matching RACMO-experiment? Polar amplification usually is strongest in winter, which is of little significance to the ice SMB.

The annual warming is indeed more pronounced than the summer warming. Based on the CMIP5 experiments (IPCC AR5), the annual warming over northern Greenland under RCP8.5 (2081-2100 vs. 1986-2005) is typically around 10°C, while the summer warming (JJA) is typically 5-7°C (50% percentile values of CMIP5 runs). As our reference level is 1961-1990, a +8°C warming can be considered as an extreme summer warming. This is now also described in the updated manuscript and we also added a reference to the IPCC's AR5 'Atlas of

Global and Regional Climate Projections' (Annex I Supplementary Material RCP8.5):
*For a high emission scenario (IPCC RCP8.5) the 2100 global average surface temperature is projected to rise by 3 to 5°C compared to the 1961-1990 average. Over high Arctic regions such as Peary Land the temperature could potentially increase by up to 7-11°C due to the polar amplification (Collins et al., 2013). This warming is most pronounced in winter, and summer temperatures (June-July-August) are projected to rise up to 8°C over northern Greenland in 2100 (vs. 1961-1990) (van Oldenborgh et al., 2013).*

**Figure 5**: Please (also) plot the difference between your modeled and the observed ice cap thickness.

An additional was plot was added (Figure 5b)

**Technical comments:**

**Page 1 – Line 14:** Please flip the direction of the comparison (SIA is the erroneous experiment) and then replace modifies with decreases (if I got the direction right in the main part of the manuscript). A new text would then be something like *Using the Shallow-Ice Approximation decreases…* I would actually prefer omitting the entire sentence, as I don't see it as relevant to your main message.

This is indeed not one of the main messages of this paper and we therefore decided to omit this sentence from the abstract.

**Page 1 – Line 22:** Please replace corresponding with a word that clearly describes causality (following?).

*Corresponding* was replaced with *resultant* to emphasize the causality.

**Page 1 – Line 26:** I think, it should read *disappear* around 2400 and 2200 A.D. **respectively**, Replace irrespective with independent (also in other locations in the text).

This was changed as suggested by the reviewer. *Irrespective* was changed to *independent*, also for the two other occurrences in the text (last sentence of section 6.1 and last paragraph of the conclusion)

**Page 3 – Line 29:** often terminate up to several hundred meters is vague in multiple ways. Also, often is with respect to time, not the individual glacier. Maybe replace with many of them terminate several hundred meters . . . ?

This sentence was modified and now reads:
*The outlet glaciers are mostly land terminating and many of them terminate up to several hundred meters above sea level*

**Page 3 – Eqn (5):** The left hand side should read $\partial_z\tau_{iz}$ ($\partial_z$ instead of $\partial_i$), e.g. eqn 5.76 of Greve and Blatter (2009)

This was modified.

**Page 6 – Line 19:** *sub/* should probably by *sub-*.

Indeed. This was changed.

**Page 6 – Line 20:** Please specify more details on the firn warming. Are we talking about firn modeled by your PDD model?

A surface warming is applied where a net annual refreezing occurs, which is only possible above the ELA, where the surface is snow/firn covered. The firn layers and their long-term evolution are not explicitly modelled in our PDD model. In order to avoid any confusion, the reference to firn was removed and the passage now reads:

*Based on field measurements (Reeh et al., 2001) a surface warming of 22 K/ m w.e. of refreezing is used.*

**Page 9 – Line 21:** please convert i.e. to w.e. (even if it means keeping the number unchanged).

This was modified → 20 K / (m i.e.) = 20 K / (0.91 m w.e.) = 22 K / m w.e.

**Page 9 – Line 22:** Please change *occurred* to *were performed* or something similar.

Changed as suggested.

**Page 14 – Line 2:** Importance **of** initial conditions

Modified.

**Page 14 – Lines 20/21:** Please rephrase *the shape of the volume evolution curve is far less exponential compared to case 1*. Maybe the growth is slower than in case 1?

We modified this:

*As a consequence of this particular ice supply, here the growth is substantially slower than in case 1 (Figure 9, 1961-1990 -0.5°C).*

**Page 15 – Line 11:** Remove *largely*. Do you mean *strongly*? Maybe provide the area change in %?

*Largely* was replaced by *strongly* as suggested and the area change in percentage is now also mentioned:

*For a cooling of only 0.5°C compared to the 1961-1990 conditions the ice cap strongly expands (21% area increase) (see Figure 9b) and the volume increases by 26%*

**Page 16 – Line 30:** I would suggest using *ΔT* instead of *T*.

*ΔT* is now being used:

*P = 0.132 ΔT² + 0.316 ΔT + 1*                                              *(11)*
*Where ΔT is the temperature forcing and P the corresponding precipitation forcing (scaling factor) (both vs. 1961-1990) needed to prevent a mass loss (vs. 1961-1990 steady state).*

**Page 17 – Line 21:** I would suggest pulling the reference to Bolch et al. (2013) to the front of the list, as it is the ice cap under investigation in this manuscript.

The order was modified as suggested:
*This is in agreement with recent ICESat observations on Arctic ice caps, which indicate a marginal ice loss and local thickening (for the interior). This is the case for the Hans Tausen ice cap (Bolch et al., 2013) and for instance also for Austfonna ice cap (Svalbard) (Moholdt et al., 2010) and the Flade Isblink ice cap (Greenland) (Rinne et al., 2011; Bolch et al., 2013).*

**Figures**

Several of the figures have a very dense raster in the background. I find it way more disturbing than helpful. Please reconsider.

The dense raster is indeed not everywhere appropriate. For figures in which a variable is plotted, the raster is kept as this helps to visually identify the exact values. For 2-D representations of a variable (i.e. a plane view visualization) the dense raster was removed (Figure 2b, Figure 7a,b, Figure 8a,b).

**Figure 1**: I found the figure hard to read. Maybe you could decrease the vertical exaggeration?

The vertical exaggeration of the figure was decreased.

**Figure 3:** (a and b) maybe a discrete colormap would make the comparison between a and b easier. This way, it is virtually impossible to read temperature values from Fig 3b. Maybe you could display RACMO downscaled with the same lapse rate as used in the analytical expression (and with bilinear/... interpolation between the grid cells)? This would make the comparison of the two plots a lot easier. Same for (c,d). Both temperature plots could (should) use the same colormap. The colormap in (c,d) very prominently marks the difference between regions with annual mean temperatures above or below -15C. Is there a specific reason for this, otherwise a linear colormap might be better.

For Fig 3 a,b,c,d a discrete colorbar is now used as suggested by the reviewer. Figure 3b and 3d were not downscaled, as we think it is important to show the model resolution here, which is one of the main reasons not to work with the RACMO temperature output directly (see also related questions above). In Fig 3c and d, there is indeed no reason to have a sharp transition at -15°C. This was now modified by removing the white component from the colormap.

**Figure 6b:** What are you telling the reader with Fig. 6b? I think, it can be omitted without loss of information.

Figure 6b gives a visual support for the 3-D calculation of the temperature field. It illustrates the change in surface and bottom temperatures with elevation and the strong contrast beween the tongue (where bed is almost at pressure melting point) and the higher parts of the outlet glacier (where basal temperatures are lower than -10°C). Furthermore it also gives a clear visualization of the ice temperature change within an ice column. We would therefore like to maintain this figure in the manuscript.

**Figure 7:** In 7 a and b you could zoom in a bit more on the ice covered domain. Then it would be displayed bigger and easier to read. I would suggest using the same color scale for the top and bottom figures. Currently one uses linear and the other one log scaling, although the data ranges seem pretty much identical. The color scale in Fig 7 c-f has large ranges where the color hardly changes at all and then sharp transitions (looks like I'd have difficulty telling 60 from 75 m/yr, but 75 to 85 is very clear). A more linear color scale (or focus on the low velocities as in a log scale) would be more helpful. The 3d-projection in c-f does not seem to add information. To me it's rather confusing. Most likely a 2d plot would do a better job.

Figure 7a and 7b were modified by zooming into the ice-covered domain. The colour scale of the figure was modified as suggested by the reviewer and now the logarithmic scale (that was originally only used for the plane view figures, a&b) is used for all figures (a-f). Our reason for opting for a 3-D projection resides in the possibility of combining topographic information with velocity information, which is not possible in a 2-D visualisation.

**Figure 8:** Please add "resolution" in the caption to make clear that 250 m and 500 m refer to the model resolution.

The figure caption was modified as suggested by the reviewer.

**Figure 9:** Again, I find the 3d-Plots very hard to read. They appear very dark. I don't think it's necessary to cite yourself again for the plot tool. The reference in Fig. 1 should be enough.

In order to improve the readability of the 3-D plots, a lighter colour scheme was used for the bedrock. The reference to the plotting toolbox was omitted.

**Figure 11:** I find Fig. 11a extremely hard to read as there is minimal contrast between the colors of lines and text and the background raster. Please change this. The same color problem applies to Fig. 11d. What is the color shading in Fig. 11c about? This should be a line plot.

The colours in Fig.11a and Fig.11d were modified in order to enhance the contrast. The colour shading in Fig. 11c is explained in the figure caption:
*The blue area broadly corresponds to the range where an attenuation of the mass loss is possible, whilst the red area represents the range under which the ice cap is to (largely) disappear*

---

## Author Comment (AC2) · 28 Feb 2017

**General comments**

This paper describes a modelling study for an ice cap in northern Greenland, outside the main ice sheet. It applies a coupled ice flow – mass balance model with PDD – runoff/retention model and precipitation downscaled from RACMO. The experiments are well designed and conclusions drawn from this study are interesting, one being that there is no gain in performing time dependent simulations for the ice cap, Holocene or future, as the response of the model is independent of the model initial conditions and there appears to be 4 sets of steady states possible for the ice cap.

The model study shows hysteresis response for a narrow band of temperature forcing (around 0.5°C higher temperature than 1961-1990 average). Authors also point out that the SMB-elevation feedback is a crucial mechanism for the ice cap evolution and stability and that infiltrating meltwater in the ablation area is necessary in order to simulate englacial temperatures. The paper is clearly written and the conclusions are clear, some minor comments for improvements are suggested below. To improve the overall clarity of the paper, better description of the model resolution as well as the temperature forcing would be good, see below.

We thank the reviewer for his positive general appreciation of the manuscript and addressed all his comments below.

**Specific comments**

It is not clear whether the input fields for the PDD scheme, precipitation from RACMO and parameterized temperatures are downscaled to the ice flow model grid resolution or if the SMB field is downscaled. Clarification in the model setup section is needed, see suggestions below. In several places it is stated that RACMO is run at 11 km and then precipitation downscaled to 1 km, but is it then further downscaled to 500 m or 250 m?

This was indeed not clear in the original manuscript. In all cases the ice flow and the surface mass balance model are run at the same resolution (250 m or 500 m depending on the experiment). In neither case is the SMB field downscaled. This has now been addressed based on the reviewer's suggestions below. The RACMO precipitation is in all cases downscaled to the model resolution. We added a sentence to clarify this in section 4.1:

*For all simulations this precipitation field is further downscaled to the model resolution*

Some confusion is in the discussion of the results, the description of the forcing is not clear, are all the forcing scenarios shifted relative to the 1961-1990 condition? And then some of them fall onto 1981-2010 or 2005-2014 mean conditions? Figure 9 has both 1961-1990 -0.5°C and different periods, but Figure 10 has all temperature scenarios relative to the reference. Some clarification in the model or experimental description is needed.

Not all forcing scenarios are shifted compared to the 1961-1990 conditions. Some of them are run under 1981-2010 conditions, while others are run with 2005-2014 conditions (with temperature and precipitation for this period). Some runs are indeed based on conditions that are shifted compared to 1961-1990 (-0.5°C and in

figure 10), in which case they have the 1961-1990 precipitation, but another temperature. The reason behind this is twofold: (i) to be able to reproduce colder conditions (cannot reproduce conditions as cold as 1961-1990 -0.5°C by taking a period of >10 years belonging to the ERA-40 period) and (ii) to allow for a continuous range of climatic conditions to be explored (Figure 10). The 1981-2010 and 2005-2014 conditions are indeed close to 'temperature shifted' 1961-1990 conditions (+0.6° and +1.6°C), but their precipitation is (slightly) different.

This has now been emphasized in the updated manuscript (see response to comments, where this addressed in detail).

**Technical corrections**

| Page 1, Abstract, Line 24, suggest to replace "grow" with "thicken" |
|---|
| This is indeed a better wording. Was modified. |

| Page 2, line 5, delete "s" on exist |
|---|
| This was modified to *exist* |

| Page 3, line 15-17, sentence is confusing, how do inconsistencies arise with bedrock from Starzer and Reeh (2001) if the direct ice thickness measurements are not included in the Bamber et al (2013) dataset? |
|---|
| The wording was not well chosen. We mean here that there are differences between the bedrock reconstruction by Bamber et al. (2013) and the one by Starzer and Reeh (2001). In the updated manuscript the sentence is now: *Notice that the direct ice thickness measurements on Hans Tausen from the 1990s are not included in the Bamber et al. (2013) dataset and therefore local differences exist with the reconstructed bedrock from Starzer and Reeh (2001)* |

| Page 6, line 31-32, make sure that the minus sign - sticks with the number |
|---|
| This is the case. In Microsoft Word '-10' is just displayed on two separate lines. But should not be a problem in final TC version (will double check it during typesetting stage). |

| Page 7, line 24, suggest to add "daily" before variability |
|---|
| This was added |

| Page 8, here the resolution of the T parameterization could be mentioned, ice flow model resolution? |
|---|
| The temperature parameterization is resolution independent. It needs the grid latitude and elevation, which can be calculated at any resolution. This is now added at the end of the first paragraph of section 4.1.2: *The temperatures are parameterized as a function of latitude and elevation, and they can therefore be determined at any model resolution.* |

**Page 9, line 11**, is the precipitation then further downscaled to ice flow model resolution?

Yes, it is. As mention above (see 'specific comments'). This was now added:

*For all simulations this precipitation field is further downscaled to the model resolution*

**Page 9, line 20**, what is "frontal SMB" - terminus ablation?

This was indeed not clearly formulated, as frontal SMB may refer to several forms of frontal ablation, including glacier calving. This was now modified:

*…for Hare glacier the SMB at the glacier terminus is…*

**Page 9, line 27-32** – here the different forcing scenarios, shifted relative to 1961-1990 or other periods, could be presented and explained

This is the model evaluation section, and introducing different forcing scenarios does not seem to be justified here. In the response to related questions this issue is further addressed (see below) and the changes in the manuscript should make this clearer.

**Page 10 line 6**, suggest to replace "further" with "below"

This was modified.

**Page 10, line 12**, what does "imposed" mean here? is the SMB regridded to 250 or 500 m resolution?

The SMB is not regridded, it is directly calculated on the 250 m resolution. In the updated version of the manuscript this will now be clearer. It is now explained that the RACMO precipitation is in all cases downscaled to the model resolution (see also the two related questions above)

**Page 10, line 21**, edit, something strange in the sentence "but except"

Indeed. These are now two separate sentences:

*…outlet glaciers are ice-covered. Except for this, the agreement is overall relatively good, especially given that there is no imposed constraint on …*

**Page 10, line 31**, is the grid not at 250m resolution?

Yes, the grid is at 250 m. But to make a direct comparison with RACMO, the SMB calculations were also performed on the GIMP 1-km grid (i.e. excluding SMB differences related to a different topography). In the new manuscript this should now be clearer. Notice that based on suggestions by Reviewer #1, this section was moved forward and changes were performed to avoid repetitions.

**Page 11, line 10**, suggest to replace "heating" with "heat flux"

This was changed.

**Page 11, line 21** and onward, it is not clear what is discussed here, paragraph needs rewriting "With a value of 150 mW m$^{-2}$…" - is not clear

This sentence was reformulated as follows:

*The modelled basal temperatures for Hare glacier are close to the pressure melting point (see Figure 6b), but nowhere basal sliding occurs*

**Page 12, lines 1-4**, strange sentence, suggest editing

This sentence was modified to:

*The observed ice thickness (Figure 2b) is well reproduced in the model (Figure 5) and so is the surface elevation (as the observed bedrock elevation is used in the model)*

**Page 12, line 17**, suggest to replace "to" with "with"

This was modified

**Page 12, line 17-20**, too long sentence, suggest to split into two

The sentence was split in two:

*For the outlet glaciers, many of the observed velocity patterns are closely reproduced. This is for instance illustrated for the main outlet glaciers at the eastern side of the ice cap…*

**Page 13, line 10**, here is seems like SMB is calculated at the ice flow model resolution, is then T and P downscaled to this resolution?

Yes, the temperature is directly calculated at the model resolution (through the elevation-latitude parameterization), while the precipitation is downscaled to the model grid. This was addressed in earlier comments by this reviewer, and the manuscript was updated accordingly (see above for more details)

**Page 13, lines 18-21**, long sentence, suggest to split or rewrite

This sentence was indeed long. It is now split in two:

*Notice that treatment of the ice mask in the downscaling approach has an important effect on the modelled geometry at a 250 m resolution. It is important that the area of the ice cap and ice-free regions is the same at both resolutions in order to ensure that the large-scale dynamics, which are determined by the overall mass balance, are similar.*

**Page 13, line 25**, "which results in smoothing…"

This was modified

**Page 14, line 2**, something is missing in title, "of"?

This was also suggested by reviewer #1 and was modified.

**Page 14, line 23**, text is confusing "(vs. " do you mean relative? Is it only temperature shift?, what are 1981-2010 conditions, do you mean in the RACMO model?

Yes, by this we mean relative to the 1961-1990 conditions. This was modified in the text. It indeed only concerns a temperature shift.

The 1981-2010 temperatures are obtained by applying a bias on the 1961-1990 temperature field (bias is based on RACMO output, which suggests that the period 1981-2010 is 0.6°C warmer than 1961-1990). Precipitation is directly taken from RACMO over the 1981-2010 period. To clarify this, an additional sentence was added:

*The 1981-2010 climatic conditions are simulated by applying a +0.6°C bias compared to the 1961-1990 temperature field (see eqs. 9 and 10), while precipitation is directly derived from RACMO2.3 for this period.*

**Page 14, line 28**, "build up faster than under 1961-1990 condition" – does not make sense, what do you mean by faster?

The sentence was changed and should now be clearer:
*Compared to 1961-1990 conditions, less time is needed for the ice cap to build up under 1981-2010 conditions, as there is no interaction between the northern and southern part of the ice cap*

**Page 15, line 5**, replace "parts" with "part"
Was modified.

**Page 15, line 15**, what does "largely similar" mean?

The 1981-2010 RACMO precipitation is very similar to the 1961-1990 RACMO precipitation. Over the Hans Tausen ice cap, the 1981-2010 mean precipitation is on 4% higher than the 1961-1990 mean precipitation. This is now also mentioned:
*(ca. +0.6°C vs. 1961-1990, 4% higher precipitation than the 1961-1990 mean)*

And further in the text also for the 2005-2014 climatic conditions:
*…which are around 1.6°C warmer than the 1961-1990 average conditions over Hans Tausen (6% higher precipitation than 1961-1990 mean), and…*

**Page 16, line 15**, needs editing, text is unclear

This sentence was modified and now reads:
*For the +4°C scenario and by maintaining the precipitation at the 1961-1990 level, the ice cap entirely disappears within 350-400 years (i.e. before 2400) (Figure 11a), disregarding whether the forcing is immediately applied at present or incrementally until 2100*

**Page 16, line 19** "slightly different transient geometry" needs editing, not clear
This was modified:
*…or a somewhat similar geometry*

**Page 17, line 20-22**, text needs editing, two times "potential" in sentence

The sentence was edited:

*An in-depth comparison between our modelling study and these observations is again not possible given the differences in timing and the model setup, but our simulations show the potential to reproduce the observed trends and the implications this can have on the future ice cap evolution.*

**Page 18, line 4**, suggest to replace "was" with "is"

This was modified.

**Page 19, line 1** "Both" – and then three things are mentioned, needs editing

This was edited by removing "both":

*Englacial temperature measurements, modelled ice thickness and temperatures in outlet glaciers suggest that there is an important heating mechanism related to infiltrating meltwater in the ablation area of the ice cap.*

**Figure 3** Annual and mean July are in different sequence in the figure and in the caption; (a), b) vs c) d)), suggest to move the precipitation figure to the right column so that RACMO fields are all in the same column

The discrepancy between the figure and the caption is well spotted by the reviewer. This was now modified. The precipitation figure was also moved to the right column (the "RACMO column").

**Figure 5**, suggest coloring dotted line white, it is hardly visible as it is now

This was modified: the line is now white and was additionally thickened.

**Figure 6 b)** is not clear, suggest less vertical exaggeration?

The vertical exaggeration of the figure was diminished in the updated manuscript.

**Figure 9** C is the period 1961-1990? Or 2010?

In this figure the steady state obtained from the 1981-2010 climatic conditions is shown (when starting from the 1961-1990 steady state geometry). To emphasize this, the fact that these are steady state geometries (9b-e) has now been added in the figure title.